# Examining protective effects of SARS-CoV-2 neutralizing antibodies after vaccination or monoclonal antibody administration

Dean Follmann [1] ✉, Meagan P. O'Brien[2], Jonathan Fintzi[1], Michael P. Fay [1], David Montefiori[3], Allyson Mateja[4], Gary A. Herman[2], Andrea T. Hooper[2], Kenneth C. Turner[2], Kuo- Chen Chan [2], Eduardo Forleo-Neto [2], Flonza Isa[2], Lindsey R. Baden[5], Hana M. El Sahly[6], Holly Janes[7], Nicole Doria-Rose [8], Jacqueline Miller[9], Honghong Zhou[9], Weiping Dang[9], David Benkeser [10], Youyi Fong[7,11,12], Peter B. Gilbert[7,11,12], Mary Marovich[1,13] & Myron S. Cohen[14]

While new vaccines for SARS-CoV-2 are authorized based on neutralizing antibody (nAb) titer against emerging variants of concern, an analogous pathway does not exist for preventative monoclonal antibodies. In this work, nAb titers were assessed as correlates of protection against COVID-19 in the casirivimab + imdevimab monoclonal antibody (mAb) prevention trial (ClinicalTrials.gov #NCT4452318) and in the mRNA-1273 vaccine trial (ClinicalTrials.gov #NCT04470427). In the mAb trial, protective efficacy of 92% (95% confidence interval (CI): 84%, 98%) is associated with a nAb titer of 1000 IU50/ml, with lower efficacy at lower nAb titers. In the vaccine trial, protective efficacies of 93% [95% CI: 91%, 95%] and 97% (95% CI: 95%, 98%) are associated with nAb titers of 100 and 1000 IU50/ml, respectively. These data quantitate a nAb titer correlate of protection for mAbs benchmarked alongside vaccine induced nAb titers and support nAb titer as a surrogate endpoint for authorizing new mAbs.

Adenovirus vector, mRNA, and protein subunit vaccines against severe acute respiratory syndrome coronavirus 2 (SARS-CoV-2) are highly effective in reducing the incidence of symptomatic infection, severe illness, hospitalization, and death due to COVID-19[1–4]. Mechanisms of COVID-19 vaccine induced neutralizing antibodies (nAbs), Fc receptor (FcR)-mediated antibody effector functions, virus-specific CD8 + T cells, and innate immune mechanisms[5–7]. nAb titer

measured shortly after the final dose of the SARS-CoV-2 vaccine series is strongly associated with protection against COVID-19 in vaccine trials[8–12]. Similarly, nAb titers are associated with protection in observational studies following natural infection, monoclonal antibody (mAb) prevention trials, and in non-human primate studies[5,13,14]. NAb titer has recently been accepted as a correlate of protection for emergency use authorization of SARS-CoV-2 variant vaccine booster

[1]Biostatistics Research Branch, National Institute of Allergy and Infectious Diseases, National Institutes of Health, Bethesda, MD, USA. [2]Regeneron Pharmaceuticals, Inc., Tarrytown, NY, USA. [3]Department of Surgery, Duke University Medical Center, Durham, NC, USA. [4]Clinical Monitoring Research Program Directorate, Frederick National Laboratory for Cancer Research, Frederick, MD, USA. [5]Brigham and Women's Hospital, Boston, MA, USA. [6]Department of Molecular Virology and Microbiology, Baylor College of Medicine, Houston, TX, USA. [7]Vaccine and Infectious Disease Division, Fred Hutchinson Cancer Research Center, Seattle, WA, USA. [8]Vaccine Research Center, National Institutes of Health, Bethesda, MD, USA. [9]Moderna, Inc., Cambridge, MA, USA. [10]Department of Biostatistics and Bioinformatics, Rollins School of Public Health, Emory University, Atlanta, GA, USA. [11]Public Health Sciences Division, Fred Hutchinson Cancer Research Center, Seattle, WA, USA. [12]Department of Biostatistics, University of Washington, Seattle, WA, USA. [13]Division of AIDS, National Institute of Allergy and Infectious Diseases, Bethesda, USA. [14]Institute for Global Health and Infectious Diseases, The University of North Carolina at Chapel Hill, Chapel Hill, NC, USA. ✉e-mail: dfollmann@niaid.nih.gov

vaccines without the additional requirement of new randomized trials to affirm clinical benefit[15,16]. The World Health Organization's (WHO's) approval of international standard methods for measuring nAb titer, including standardization across platforms, allows nAb immuno-bridging across COVID-19 vaccines[17].

Antiviral mAbs are effective in the treatment and prevention of COVID-19[18–23] and are important for immunocompromised and medically frail populations who respond poorly to vaccines[18,20,21,24,25]. SARS COV-2 is rapidly evolving with global emergence of variants of concern (VOC) and variants of interest (VOI), including the latest variants BQ.1, BQ.1.1, XBB, and XBB.1, which are all derived from Omicron lineage. Increasingly, SARS CoV-2 variants are evading vaccine-derived and infection-derived immunity[26,27], as well as evading binding and neutralization by available antiviral mAbs[28]. Yet, unlike vaccines, authorization of preventative mAbs targeting VOC appears to necessitate increasingly difficult to conduct clinical trials requiring clinical endpoints[29]. A nimbler assessment of next generation mAbs is urgently needed to provide up-to-date preventive and therapeutic options for immunocompromised individuals.

Clinical efficacy data from a randomized prevention trial (COV-2069) of the mAb combination, casirivimab, and imdevimab, both highly potent neutralizing antibodies recognizing the spike protein of SARS-CoV-2[18,25] and the Coronavirus Efficacy (COVE) trial of the mRNA-1273 vaccine were analyzed together with representative blood samples from the respective studies using a validated lentivirus-based pseudo-virus neutralization assay to estimate the relationship between serum nAb titers over time and preventive efficacy against symptomatic COVID-19 illness (COVID-19) in SARS-CoV-2 naïve (seronegative) individuals. We contrasted the mRNA-1273 vaccine and casirivimab + imdevimab mAb curves of protective efficacy as a function of nAb titer at exposure to SARS-CoV-2 and assessed the role of nAb titer in vaccine induced protection as compared to mAb protection. We also quantitated nAb titers in mAb induced protection from COVID-19 to assess potential use of nAb titer as a correlate of disease prevention.

## Results
### Neutralization titer kinetics
For participants in the monoclonal antibody COV-2069 trial, the median half-life of the neutralization titer was 26 days. For vaccinated individuals in the COVE trial, the estimated half-life was 70 days. The predicted nAb kinetics of 10 randomly selected COV-2069 and COVE participants over the course of their respective follow-up periods are given in Fig. 1.

### Monoclonal antibody (casirivimab + imdevimab) preventive efficacy
The cumulative incidence of COVID-19 by randomization group is given in Supplementary Fig. S1A starting at day 8 post injection. Over the course of 8-months of follow-up, and excluding the first week, there were 11 cases of COVID-19 in the casirivimab + imdevimab arm compared to 63 in the placebo arm resulting in an estimated PE of 100% x (1−11/63) = 82.5%. Over the three intervals post injection, 8–29 days, 30–164 days, and 165–219 days, the casirivimab + imdevimab/ placebo case ratios were 3/22, 0/33, and 8/8, respectively, which correspond to crude PEs of 86%, 100%, and 0%.

### Vaccine (mRNA-1273) efficacy
Starting 7 days after the day 57 visit through the end of the blinded phase, a total of 47 vaccine group participants and 659 placebo group participants acquired COVID-19 resulting in an estimated VE of 100% x (1−48/659) = 92.6%. The cumulative incidence of COVID-19 by arm is given in Supplementary Fig. S1B and shows no evidence of waning efficacy.

### Efficacy curves for the monoclonal antibody combination (casirivimab + imdevimab) and mRNA-1273 vaccine
The relationship between predicted log10 nAb titer at the time of exposure and protection against COVID-19 is shown in Fig. 2. For monoclonal antibody-related efficacy associated with casirivimab + imdevimab use there is high estimated PE with relatively narrow confidence intervals for higher titers and large uncertainty for lower titers due to relatively few COVID-19 cases. Predicted nAb titer was significantly correlated with protection ($p < 0.01$, likelihood ratio test statistic 10.12 on 2 degrees of freedom). NAb titers of 1000 IU50/ml of the mAb combination were associated with a PE of 92% (95% CI: 84%, 98%). Efficacy greater than 80% was achieved with a nAb titer of 398 IU50/ml (95% CI: 25 IU50/ml, 631 IU50/ml), the maximal efficacy was estimated as 0.94 (95% CI: 0.85, 0.99), and the titer associated with 50% PE was 200 IU50/ml (95% CI: 5 IU50/ml, 501 IU50/ml). Similar curves with lower efficacy are estimated for any SARS-CoV-2 infection

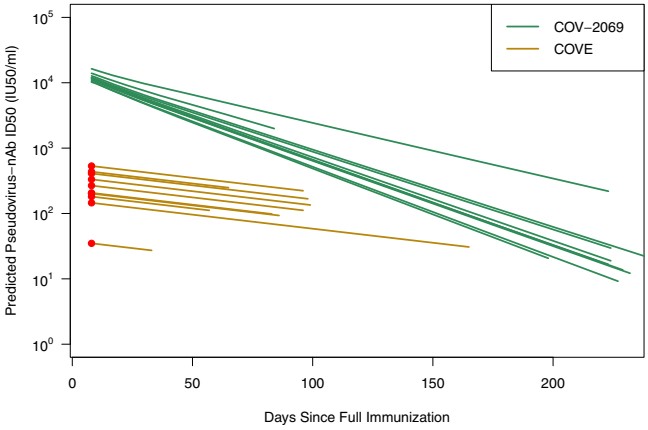

**Fig. 1 | Predicted pseudo-virus neutralization titer by days since full immunization (day 1 for mAbs, day 57 for vaccine) for ten randomly selected participants from the COVE immunogenicity subcohort (gold) and COV-2969 (green) trials.** The COVE lines use the measured Day 57 neutralization titer (red circle) with subsequent decay determined by a common slope estimated from independent data. Casirivimab + imdevimab mAb lines use concentration curves based on sex and weight and subsequently converted to neutralization titer. The curves start at day 8 days post full immunization (vaccine) or injection (mAb) and stop at the time of event or the end of follow-up.

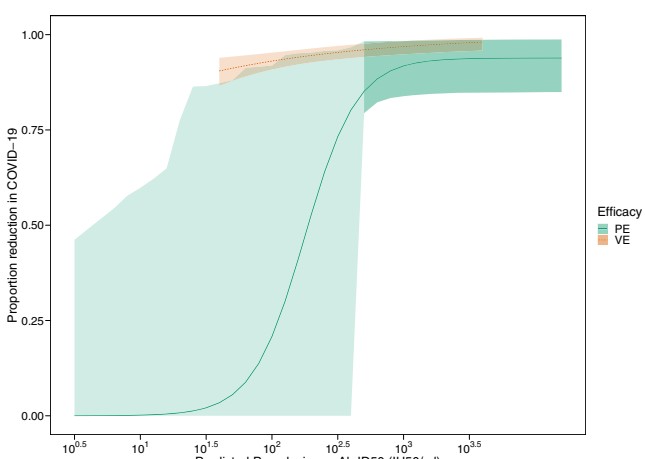

**Fig. 2 | Protective efficacy (PE) of casirivimab + imdevimab mAbs (solid green curve) and vaccine efficacy (VE) of mRNA-1273 (dashed orange curve) against COVID-19 as a function of predicted pseudo-virus neutralization titer at the time of exposure.** Shaded area provides 95% pointwise confidence intervals with lighter green emphasizing greater uncertainty for lower titers. PE and VE curves cover the distribution of titers achieved during follow-up with no extrapolation.

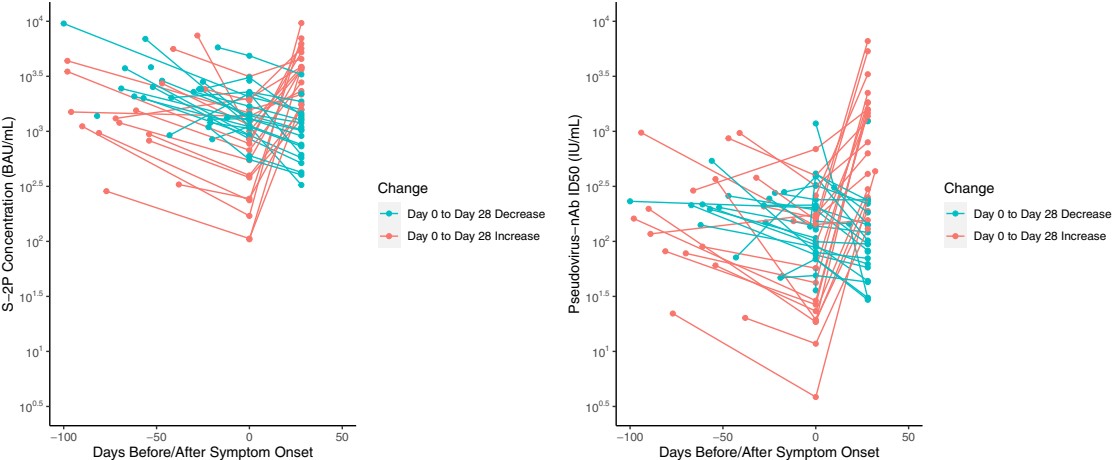

**Fig. 3 | Binding antibody concentration and neutralization titers at 28 days post second dose, the onset of COVID-19 symptoms, and 28 days later in vaccinated participants who acquired COVID-19 during the blinded phase of COVE.** Day 0 is the onset of symptoms.

and asymptomatic infection (Supplementary Figs. S2 and S3). Qualitatively similar results with tighter confidence intervals are estimated with a log-linear PE curve using Cox regression (Supplementary Fig. S4). For the mRNA-1273 vaccine, predicted nAb titer was also significantly correlated with protection ($p < 0.01$, likelihood ratio test statistic 8.20 on 1 degree of freedom). The VE curve has narrow confidence intervals due to many more cases of COVID-19 than COV-2069; and neutralization titers of 100 and 1000 IU50/ml were associated with VEs of 93% (95% CI: 91%, 95%) and 97% (95% CI: 95%, 98%), respectively, demonstrating high mRNA-1273 vaccine efficacy for a wide range of nAb titers. A bootstrap test of the equality of the PE and VE curves at a titer of 40 IU50/ml, the lower limit of titers achieved during COVE follow-up, rejects at $p < 0.05$. A generalized Wald test of equality of the log-linear PE and VE curves in Fig. S4 rejects at $p < 0.001$.

### Quantifying the role of extant nAb titers in vaccine induced protection

We deconstructed the total vaccine effect as the product of an extant nAb titer effect times all other vaccinal effects beyond extant nAb titer. The analysis suggests that at a predicted neutralization titer of 1000 IU50/ml, the percent of total vaccine efficacy due to extant antibody was 72% (95% CI: 50%, 100%) and the probability that a person protected by the vaccine would have been protected by the mAb at the same titer level was 0.95 (95% CI: 0.86, 1.00) (Supplementary Sections 1 and 2). For lower nAb titers, mRNA-1273 vaccine efficacy remains high while casirivimab + imdevimab mAb efficacy declines, however, an accurate quantitative deconstruction of mRNA-1273 vaccine protective efficacy is precluded by the high uncertainty due to few COVID-19 events at lower nAb titers.

### In mRNA-1273 vaccine disease cases nAb titers rise in those with lower titers

To further investigate the effect of an anamnestic versus extant antibody response in protection, we evaluated the kinetics of antibody levels in vaccinated disease cases during the blinded phase of the COVE trial[30]. Fig. 3 graphs both binding and neutralizing antibody levels 28 days after the 2nd dose, at the onset of symptoms, and 28 days later. Based on random effects modeling, the half-lives of antibody abundance from 28 days post 2nd dose to the onset of symptoms was estimated as 58 days for binding antibody and 62 days for neutralization antibodies. We identified those who had antibody measurements at both the onset of symptoms (Day 0) and 28 days later and calculated the median antibody at day 0. We then split the participants into two groups whose Day 0 antibody magnitude was above or below this median. The average rise in spike log10

concentration (BAU/ml) was 0.53 for the low group and 0.07 for the high group ($p = 0.015$ using a $t$-test). Results were similar for neutralization titer with the average log10 ID50 titer rising 0.76 for the low group and 0.13 for the high group ($p = 0.018$ using a $t$-test). These results suggest that for higher titer vaccinated individuals, symptomatic infection can be cleared without invoking a measurable anamnestic response. We speculate that for asymptomatic infections that do not develop into COVID-19, an anamnestic response may be important at lower titers, but not at higher titers.

### Discussion

Although the mRNA-1273 vaccine and casirivimab + imdevimab mAbs for COVID-19 prevention have high efficacy over the duration studied, the relative contribution of antibody at the time of exposure differs. For nAb titers greater than 1000 IU50/ml, the clinical efficacy of both mRNA-1273 and casirivimab + imdevimab to prevent COVID-19 is >90%, with extant antibody being responsible for approximately 72% of the total vaccine effect at a nAb titer of 1000 IU50/ml. At lower titers (e.g., <100 IU50/ml), mRNA-1273 efficacy persists, in contrast to the waning efficacy associated with the mAb combination casirivimab + imdevimab, where precise quantitation of protective efficacy at lower nAb titers was not possible due to the low number of COVID-19 cases at late timepoints in COV-2069.

While the efficacy of casirivimab + imdevimab to prevent SARS CoV-2 is driven entirely by passive immunity with exogenous monoclonal antibodies[30], the mechanisms of vaccine induced protection are more complex. Vaccination induces polyclonal antibodies as well as memory B cells, and T cells. In observational studies, nAbs protect against subsequent infection and mitigate the severity of COVID-19[31,32] and neutralization levels are highly predictive of protection across several vaccine and natural infection cohorts[11]. In a case-cohort study of mRNA-1273 breakthrough infections, day 57 anti-spike IgG concentrations and nAb titers were inversely correlated with the risk of COVID-19 infection[8]. However, non-human primate studies and clinical studies have illuminated the potential role for cellular immunity, including memory B cell responses and T cell responses[5,33]. Vaccine efficacy is likely driven primarily by nAbs and anamnestic antibody responses but also strengthened by T cell responses, which aid in protection against severe COVID-19, especially in the case where SARS-CoV-2 variants escape nAb responses[5,33–35]. Our data indicate a role for vaccine effects beyond extant antibody especially at low neutralization titers both because VE remains high while PE wanes, and because disease breakthrough cases at low titers induce an anamnestic response which is largely absent at higher titers. In aggregate, our results support extant antibody as a mechanistic correlate of

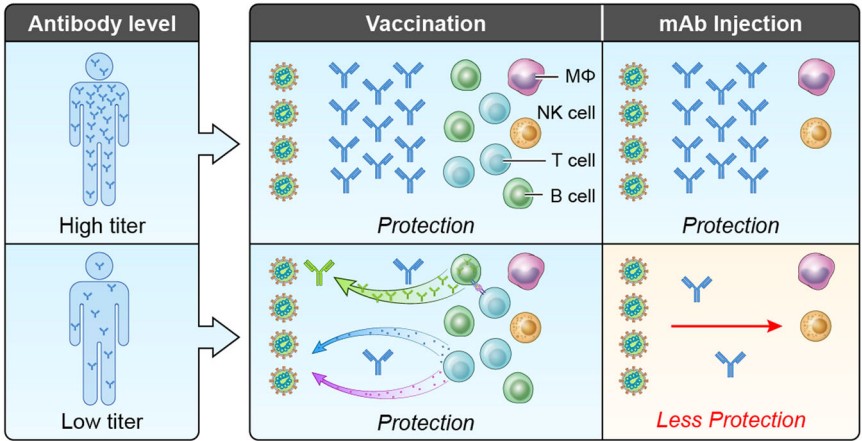

**Fig. 4 | A schematic illustrating the role of extant circulating and possibly mucosal antibodies in vaccine induced protection.** Four identical exposure scenarios are depicted. At a higher neutralization titer e.g., 1000 IU50/ml, extant antibody alone results in protection with no need for engagement of B cells, T cells, or other vaccine induced elements. At a lower titer e.g., <100 IU50/ml, mAb antibody alone is not enough for high protection. Thus, vaccine induced protection for lower titers requires engagement of some combination of anamnestic B cell responses (e.g., nAbs antibodies that mediate FcR effector functions), CD8 + T cells, and other vaccine induced immune responses.

protection in humans, whose relative importance may vary with the titer level at exposure. Figure 4 gives a schematic summarizing this interpretation for vaccine and mAb participants with low and high titers at the time of exposure to SARS-CoV-2.

Vaccine induced nAbs are highly correlated with the prevention of COVID-19 in multiple vaccine trials[5,8] and meta-analyses[9,11]. Both the Food and Drug Administration and the International Coalition of Medicines Regulatory Authorities now recommend the use of vaccine induced nAb titer as a correlate of protection for the authorization of new variant vaccines and booster doses based on immunogenicity studies[15,16] of historical clinical trials with clinical endpoints in the setting of ancestral or early VOCs.

The European Medicines Agency recently recommended authorizing Sanofi's monovalent B.1.351 protein vaccine as a booster based on clinical immunobridging with a BNT162b2 mRNA comparator vaccine using nAb titer as an endpoint.

In contrast, authorization of mAbs appears to require clinical endpoint studies, even though the case for nAb titer as a surrogate of protection is mechanistically stronger as the mAb is the single component of the intervention. Furthermore, recent meta-analyses show similar relationship between nAb titer and protection for vaccines and the monoclonal antibody adintrevimab[13,14]. For prevention of HIV, mAb titer has been proposed as a surrogate endpoint for evaluation of new mAb cocktails using methods similar to ours[36].

The current approach to Food and Drug Administration (FDA) and Emergency Use Authorization (EUA) regulation of mAbs is challenging. While nAb titer alone is currently insufficient as a surrogate of protection to support EUA, nAb titers are monitored against VOCs and inform recommendations to discontinue the use of previously authorized mAbs if there is evidence of substantial reduction in nAb titers to emergent variants. For example, the combinations balmlanivimab + estesvimab, casirivimab + imdevimab and sotrovimab are no longer authorized for treatment of COVID-19 based on low neutralization activity against the Omicron-lineage VOC. Evusheld™ remains available for prevention of COVID-19 based on the activity of one of the mAbs to neutralize the Omicron VOC, although the dosage approved was doubled as new VOCs arose. Thus, nAb titer is being used as a surrogate endpoint against emerging VOCs for established mAbs but not for the development of new mAbs. This inconsistency in the use of neutralization titer led to a recommendation for the WHO to update their Living Guidelines on mAbs and to consider using neutralization titer in conjunction with efficacy data for regulatory

purposes[29]. Moreover, none of the developed mAbs retain any neutralizing activity against BQ.1 or BQ1.1, now dominant in the U.S and other geographies, raising the specter of a long gap where no monoclonal antibodies for prevention or therapy are available for the immunocompromised and elevating the sense of urgency for a reappraisal of approaches.

Our study has the following limitations: mRNA-1273 induced nAbs declined relatively little during follow-up and the correlate of protection curve was not estimated for low titers, therefore not allowing for a side-by-side comparison of mRNA-1273 and casirivimab + imdevimab induced nAbs at low titers. Our studies were conducted in the pre-Omicron era limiting the generalizability of our results to additional variants. Ongoing work is assessing the impact of neutralization titer on Omicron disease and will be important to contrast with our results. While the COVE follow-up ended before the emergence of Delta, COV-2069 did extend into the Delta era. In deconstructing the role of extant antibody in vaccine induced protection, we used casirivimab and imdevimab which are both IgG1 antibodies and are an imperfect proxy for mRNA-1273 induced antibodies which are polyclonal and include other IgG isotypes, IgM and IgA, which may differ in their potential to penetrate the mucosa and in non-neutralizing functions. Our analyses are for a specific mAb combination and vaccine, similar analyses for other mAbs and vaccines would be informative. In the COV-2069 trial, testing for asymptomatic infection was participant driven after the first month resulting in relatively few asymptomatic cases later in the trial. We used predicted neutralization titer for each individual throughout follow-up; using actual titers from frequent sampling might sharpen our results but is not logistically feasible. Baseline characteristics of the participants in the two trials differ somewhat, ideally a randomized 3 arm trial of placebo, mAb, and vaccine would be analyzed. In the COV-2069 trial some vaccination occurred which might weaken our results. Finally, while the confidence of protective efficacy of casirivimab + imdevimab at titers >1000 IU50/ml is high, there is large uncertainty about the extent of protective efficacy at lower titers.

Using clinical and drug concentration data from a randomized, controlled prevention trial with casirivimab and imdevimab (completed prior to the emergence of Omicron and Omicron lineage VOCs), alongside ex vivo data from a standardized neutralization assay using reference strain D614G, we identified a strong correlation between mAb neutralization titer and protective efficacy. This result coupled with evidence from other mAb trials, acceptance of neutralization titer against variants for established mAbs, and the extensive evidence from

vaccine studies strongly supports the consideration of neutralization titer as a surrogate of clinical efficacy for the development of next generation mAbs for emerging SARS-CoV-2 variants. As SARS-CoV-2 evolution compromises the benefits of available mAbs, this issue has great urgency for millions of immunocompromised hosts who respond poorly to vaccines.

## Methods

### Study design and population: casirivimab + imdevimab mAb prevention trial

Details of the casirivimab + imdevimab COV-2069 clinical trial for the prevention of COVID-19 are described elsewhere[25]. Briefly, in the trial designed to assess both post- and pre-exposure prophylaxis, participants (≥12 years of age) were randomized in a 1:1 ratio within 96 h after a household contact was diagnosed with a SARS-CoV-2 infection to receive a total dose of 1200 mg of casirivimab and imdevimab (600 mg each) (henceforth mAb arm) or placebo, administered subcutaneously. At the time of randomization, participants were stratified according to the results of the local diagnostic assay for SARS-CoV-2, if available, and age. The trial consisted of a screening–baseline period, a 28-day efficacy assessment period (EAP) to assess post- and pre-exposure prophylaxis, including a weekly nasopharyngeal swab for central laboratory SARS CoV-2 RT-qPCR testing and weekly interview for symptoms assessment, and a subsequent 7-month follow-up period to assess pre-exposure prophylaxis (totaling 8-months of follow-up). Throughout the 7-month post-EAP follow-up period, participants who were symptomatic underwent assessment and central laboratory RT-qPCR testing of nasopharyngeal swabs; if the participant was unable to have a central laboratory RT-qPCR test, local laboratory molecular testing was used. COVID-19 was defined as symptomatic RT-qPCR-confirmed SARS-CoV-2 infection using a broad list of potential symptoms that would constitute COVID-19, as previously described (See Supplementary Section 3). Participants were also assessed for asymptomatic infection by RT-qPCR, weekly through the 28-day EAP, and then via participant driven testing in the case they were screened for school, work, or close contact exposure in their community, according to local guidance, etc. We analyze asymptomatic infection in supportive analyses. The trial was conducted at 112 sites in the United States, Romania, and Moldova between 13 July 2020 and 4 October 2021 prior to the emergence of Omicron-lineage VOC. Because the median time from exposure to onset of symptoms was approximately 4–5 days[37], cases accruing within the 1st week were not included. The analysis set included participants without evidence of SARS CoV-2 infection at baseline and whose illness (signs and symptoms of COVID-19 with PCR confirmation) began after the 1st week of the study to assess the preventative effect of casirivimab + imdevimab. The median follow-up was 225 days.

The analysis set of participants without SARS-CoV-2 through day 8, included 829 and 801 participants in the mAb, and placebo arms, respectively (Supplementary Table S1) While vaccines were prohibited prior to randomization, 35% of participants reported receiving at least one dose during the follow-up period with a median of 109 days to first dose and were included in our modified intent-to-treat (ITT) analysis set. Three participants (two on placebo) developed symptomatic infection after vaccination.

The mAb and placebo arm participants display similar characteristics with a mean age of 42 years, 47% male, 86% white, mean body mass index (BMI) of 29, and 11% health care worker or first responder.

### Study design and population: COVE mRNA-1273 vaccine trial

Details of the COVE trial through the end of the blinded phase are published elsewhere[1,8,38]. Briefly, COVE was a double-blind, randomized, placebo-controlled evaluation of the efficacy, safety, and immunogenicity of the mRNA-1273 SARS-CoV-2 vaccine in participants who were ≥18 years old and had no known history of SARS-CoV-2 infection and were at appreciable risk of acquiring SARS-CoV-2 infection or high-risk for severe disease. Participants were randomly assigned in a 1:1 ratio to receive two doses (28 days apart) of the mRNA-1273 vaccine (100 μg) or placebo and with randomization stratified according to age and COVID-19 complications risk criteria (i.e., ≥18 to <65 years and not at risk, ≥18 to <65 years and at risk, and ≥65 years). At day 57 post first dose (28 days post second dose), participants had blood drawn for immune correlates analyses. COVID-19 was defined similarly to the casirivimab + imdevimab trial as symptoms plus a positive PCR test (See Supplementary Section 3). The trial was conducted at 99 sites in the United States with blinded follow-up between 27 July 2020 and 26 March 2021 and the median follow-up was 157 days.

Gilbert et al[8]. conducted an immune correlates analysis based on the COVE trial through the end of the blinded phase using day 57 antibody measurements in participants with no evidence of infection at baseline through 6 days post Day 57 visit, where COVID-19 cases were counted starting day 7 days after Day 57. Neutralization titer was measured against the D614G strain. Day 57 neutralization titers of 100 and 1000 IU50/mL were associated with cumulative vaccine efficacies of 91% and 96%, respectively. Here, we use the same data as Gilbert et al.[8], but correlate the incidence of COVID-19 events with predicted neutralization titers throughout follow-up, rather than at day 57, thus analyzing titer as an exposure proximal correlate. In the analysis set of participants without SARS-CoV-2 through day 63, 14142 were in the mRNA-1273 arm and 13906 in the placebo arm. The two arms displayed similar characteristics with a mean age of 52 years, 52% male, 80% white, a mean BMI of 29, and 27% that were health care worker or first responders (Supplementary Table S2).

We also used data from Follmann, Janes et al., who evaluated the kinetics of the antibody response to disease during the blinded phase of COVE, for supportive analysis[30]. For vaccinated disease cases and matched placebo disease cases, both binding antibody to spike and neutralization titers were measured during the primary immunization series, at the onset of symptoms, 28 days later, and at the end of the blinded phase. In vaccinated disease cases there was a modest increase in antibody 28 days after symptom onset, in sharp contrast to the placebo disease cases who had a substantial response similar to a single dose of mRNA-1273.

A schematic of the two studies is given in Fig. 5. Both studies were conducted before the emergence of Omicron-lineage VOC in participants in the trials, had similar populations, enrolled SARS-CoV-2 naïve individuals, and had similar definitions of symptomatic COVID-19 illness (See Supplementary Section 3).

### Neutralizing antibody data

For COV-2069 trial participants, pharmacokinetic (PK) antibody concentration curves of casirivimab and imdevimab in serum over 8-months were estimated using population PK models developed for casirivimab and imdevimab from three clinical studies. The population PK models were two compartment models with linear elimination and first-order absorption following subcutaneous dosing[25]. Concentrations of casirivimab and imdevimab at each time point were added to obtain concentrations of casirivimab and imdevimab combined in serum. Stochastic simulations with inter-individual random effects were used to predict casirivimab and imdevimab concentration by weight and sex for each day of follow-up. Figure 6A displays the concentration time profiles for 10 randomly selected individuals. To transform antibody concentration to pseudo-virus neutralization titer, we conducted a separate experiment where 27 frozen serum samples from COV-2069 participants that spanned the course of follow-up were selected. Eighteen samples had both concentration measured and neutralization titer above the limit of detection. We estimated the relationship between the concentration of antibodies and the neutralization titer for these 18 samples as $\log_{10}(ID50) = 1.80 + 1.15 \times \log_{10}(\text{concentration})$, Fig. 6B. We used this equation to determine

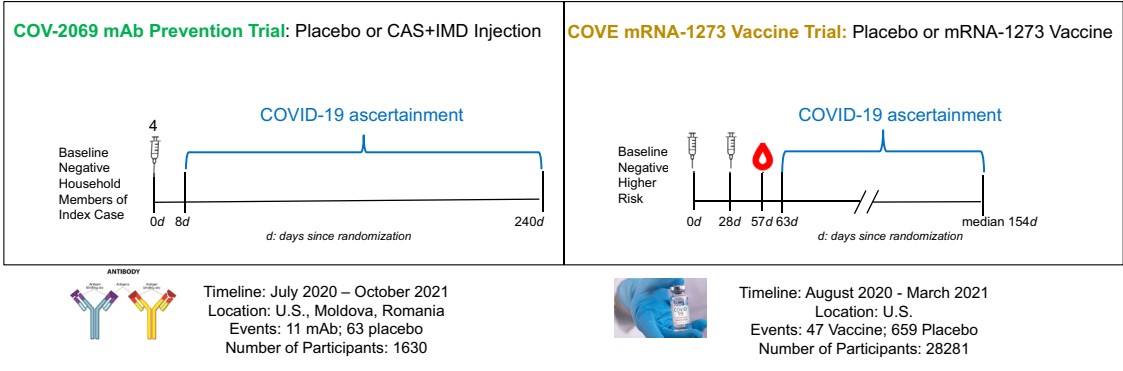

CAS + IMD: casirivimab and imdevimab; mAb: monoclonal antibody; PCR: polymerase chain reaction

**Fig. 5 | Schematic of the COV-2069 mAb prevention trial and mRNA-1273 vaccine trial (COVE) as analyzed in this report.** Both trials were conducted before the emergence of the Omicron VOC in trial participants, had similar populations, enrolled SARS-CoV-2 naïve individuals, and had similar definitions of symptomatic COVID-19 illness.

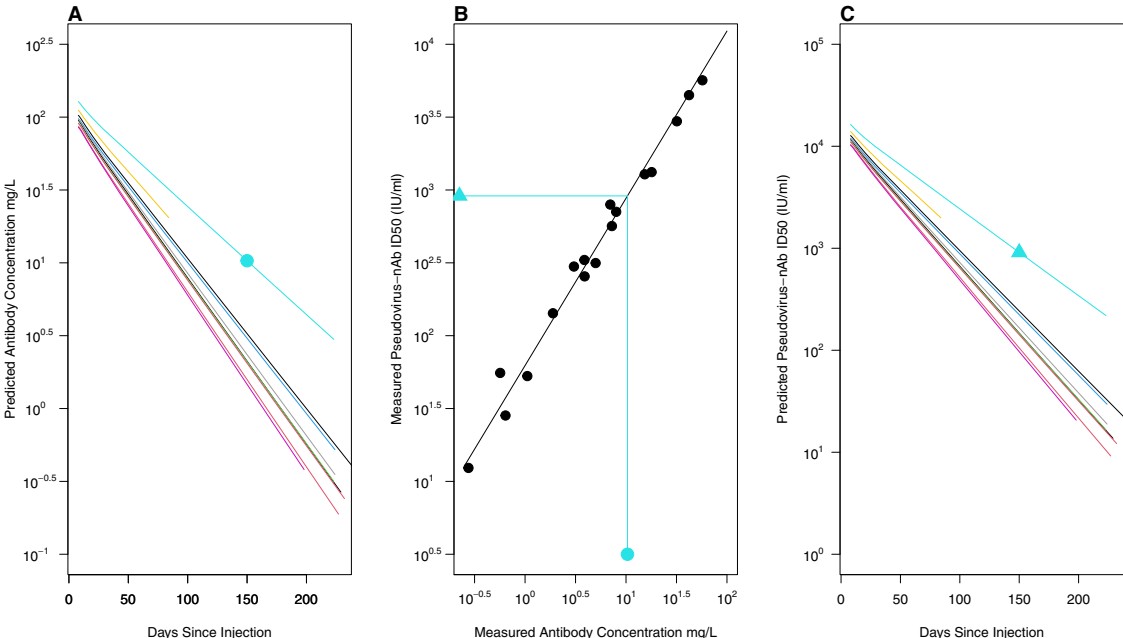

**Fig. 6 | Derivation of the pseudo-virus neutralization titer decay curves for casirivimab + imdevimab mAb combination. A** Ten randomly selected population PK predicted casirivimab + imdevimab mAb combined antibody serum concentration vs time curves as a function of days since injection. **B** 18 paired measurements, sampled throughout the course of follow-up (black dots) were used to estimate a linear relationship (black line). A day 150 population PK predicted mAb concentration (turquoise dot) thus results in a predicted ID50 (turquoise triangle). The predicted ID50s were used to generate neutralization titer decay curves (Panel **C**).

individualized neutralization titer decay curves (Fig. 6C). The process is illustrated by the turquoise line and symbols where a log10 concentration of 1.01 mg/L on day 150 post injection is transformed to a log10 neutralization titer of 2.96 IU50/ml.

For COVE, 36 vaccinated COVID-19 cases starting at least 7 days after day 57 were added to the immunogenicity subcohort (a stratified random sample of participants augmented with all cases) and had D614G pseudo-virus nAb titer measured on day 57[8]. Individualized log-linear neutralization titer decay curves were constructed by adjusting the day 57 titer by a decay factor that was estimated using independent data from ref. [39], where 34 participants had a pseudo-virus nAb titer measured on 57 (peak immune response), 119, and 209 following the first dose of vaccine (See Supplementary Fig. S5) using a SARS-CoV-2 D614G spike-pseudotyped virus[39]. In a further separate experiment,

68 paired samples were used to demonstrate a high concordance between the pseudovirus assay used herein (Duke) and the pseudo-virus assay performed by Doria-Rose et al. at the NIH Vaccine Research Center (VRC)(See Supplementary Section 4 and Fig. S6).

Neutralization titers are reported in international units as IU50/ml = ID50 x 0.242 where ID50 is the reciprocal ID50 dilution titer (see Gilbert et al.[8], and Supplementary Section 5). These titers in IU50/ml can be converted to reciprocal dilution titers by dividing by 0.242. Thus, a titer of 1000 IU50/ml corresponds to an ID50 of 4132.2.

## Statistical analyses
For each individual in each trial a predicted neutralization titer on each day throughout follow-up was constructed as follows. The relationship between log10 mAb concentration of casirivimab + imdevimab and

log10 pseudovirus neutralization titer was estimated using linear regression. For each participant in the COV-2069 trial, the slope from a linear regression of predicted log10 neutralization titer on days since injection was estimated over 8-months of follow-up. For the COVE antibody kinetics analysis, a hierarchical Bayesian model was used to estimate the posterior distribution of the rate of decay of the log neutralization titer, denoted B, over days 57, 119, and 209. The log10 neutralization titer on any day $d$ post day 57, was predicted as the day 57 value plus B x $d$ (See Supplementary Section 6). The relationship between the Duke and VRC pseudo-virus neutralization assays was estimated using Deming regression. Proportional hazards regression models were used to assess the instantaneous risk of COVID-19 as a function of predicted log10 neutralization titer throughout follow-up and to derive vaccine and protective efficacy functions. For COVE, a log-linear curve was used

$$VE(Ab) = 1 - \exp(\beta_0 + \beta_1(Ab)), \tag{1}$$

where Ab is log10(ID50) (See Supplementary Section 7). Log linear VE curves were used in a prior analysis of COVE and multiple vaccine trials thus allowing comparisons with other studies[40]. For COV-2069 a three-parameter logistic curve was used

$$PE(Ab) = 1 - \{\theta + (1 - \theta)\text{expit}(\beta_0 + \beta_1 Ab)\} \tag{2}$$

Where expit(a) = exp(a)/(1+exp(a)) (See Supplementary Section 8). This curve allows for flexible modeling of a wide range of PEs. The maximal protective efficacy is given by $(1 - \theta)$, the ratio $-\beta_0/\beta_1$ determines the level of Ab where the maximal protective efficacy of $(1 - \theta)$ is halved, and as this ID50 goes to 0, PE goes to zero. We also estimated a log-linear curve for PE as a sensitivity analysis. Likelihood ratio tests were used to assess whether protection varied with predicted neutralization titer. A bootstrap approach was used to test equality of the PE and VE curves at specific titers and a generalized Wald test used to test overall equality of the log-linear VE and PE curves. The fraction of the total vaccine effect due to nAb titer and the probability of mAb protection given vaccine protection, i.e., deconstruction analysis, is described in Supplementary Section 1. For each trial the bootstrap percentile method with 10,000 bootstrap samples was used to provide confidence intervals for the estimated parameters and efficacy curves and to test equality of the VE and PE curves. For each bootstrap iteration, predicted neutralization titers over follow-up were generated from a parametric bootstrap based on the estimated titer decay model (See Supplementary Section 9 and Fig. S7 for model goodness-of-fit testing). No sex or gender analyses were conducted. All statistical tests are two-sided. Analyses were conducted using R 4.1.0 and 4.2.0. Details of all statistical methods are provided in Supplement Sections 1, 2, 4–9).

**Reporting summary**

Further information on research design is available in the Nature Portfolio Reporting Summary linked to this article.

## Data availability

For COV-2069

Qualified researchers can request access to study documents (including the clinical study report, study protocol with any amendments, blank case report form, and statistical analysis plan) that support the methods and findings reported in this manuscript. Individual anonymised participant data will be considered for sharing once the product and indication has been approved by major health authorities (e.g., US Food and Drug Administration, European Medicines Agency, Pharmaceuticals and Medical Devices Agency, and so on), if there is legal authority to share the data and there is not a reasonable likelihood of participant re-identification. Requests should be submitted to https://vivli.org/. Regeneron does not commit to a specific timeframe to respond to requests. Requests are vetted in the order that they are received. As for the question regarding restrictions, all data requestors must execute Vivli's Data Use Agreement https://vivli.org/wp-content/uploads/2022/06/2022_06_21-Vivli-Data-Use-Agreement-v1.3.pdf before access is granted. The agreement is publicly available at the link provided. Additionally, the requirement to sign the contractual agreement is noted in REGN's publicly available data sharing policy which can be found here: https://www.regeneron.com/downloads/clinical-trial-disclosure-data-transparency-policy.pdf#:~:text=Regeneron%20is%20committed%20to%20sharing%20clinical%20trial%20data,a%20Regeneron%20sponsored%20study%20by%20submitting%20a%20research

For COVE

As the trial is ongoing, access to participant-level data and supporting clinical documents with qualified external researchers may be available upon request and is subject to review once the trial is complete. Such requests can be made to Moderna Inc., 200 Technology Square, Cambridge, MA 02139, USA. A materials transfer and/or data access agreement with the sponsor will be required for accessing of shared data. All other relevant data are presented in the paper. The protocol is available in the Supplementary Information: Clintrials.gov. NCT04470427. Data requests should have a response within 2 weeks.

## Code availability

Code to estimate the proportional hazards models used in the manuscript are available on github https://github.com/follmand/COVID-19-preventative-efficacy-correlated-with-neuts.

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

## Acknowledgements

We thank the volunteers who participated in the COV-2069 and COVE clinical trials. We thank Heather Angier for assistance with technical editing. This project has been funded in whole or in part with federal funds from the National Cancer Institute, National Institutes of Health, under Contract No. 75N91019D00024. The content of this publication does not necessarily reflect the views or policies of the Department of Health and Human Services, nor does mention of trade names, commercial products, or organizations imply endorsement by the U.S. Government. These studies were supported by the National Institutes of Health/National Institute of Allergy and Infectious Diseases. The content is solely the responsibility of the authors and does not necessarily represent the official views of the National Institutes of Health.

## Author contributions

Conceptualization: D.F., M.M., and M.F. Methodology: D.F., M.F., and J.F. Formal Analysis: D.F., M.F., J.F., and A.M. Writing – original draft: D.F., M.C., M.O.B., and M.M. Writing – review & editing: D.F., J.F., M.P.F., D.M., A.M., H.M.E.S., H.J., N.D.-R., D.B., Y.F., M.M., M.P.O., G.A.H., A.H., K.-C.C., E.F.-N., K.C.T., F.I., L.R.B., J.M., H.Z., W.D., P.G.B., and M.S.C.

## Funding

## Competing interests

D.F., J.F., M.P.F., D.M., A.M., H.M.E.S., H.J., N.D.-R., D.B., Y.F., and M.M. have no competing interests to declare. M.P.O. is an employee, has stock options, a patent pending, and license and royalties with Regeneron Pharmaceuticals, Inc. G.A.H. is an employee and shareholder of Regeneron and is listed on pending patents for the REGEN-COV antibody cocktail. A.H. owns stock in Regeneron and Pfizer. K.-C.C. and E.F.-N. are employees and shareholders of Regeneron. K.C.T. and F.I. are employees and shareholders of Regeneron and are listed on a pending patent. L.R.B. is the Deputy Editor for the New England Journal of Medicine and has grants from the Bill and Melinda Gates Foundation, Harvard Medical School, the National Institutes of Health, and the Wellcome Trust. J.M. is an employee of and has stock options and stock grants from Moderna. H.Z. is an employee of and has stock options from Moderna. W.D. is an employee of Moderna. P.G.B. will be serving as an unpaid advisor on Moderna's Zika Vaccine Advisory Board. M.S.C. serves on the scientific advisory boards of Aerium, ModexX, and Atea and has consulting roles with Astra Zenica and GSK.

## Ethics approval

We have complied with all relevant ethical regulations in analyzing these data. For COV-2069 the central or local institutional review board or ethics committee at each study center oversaw trial conduct and

documentation. The central IRB was WCG-IRB For COVE a central institutional review board, Advarra, approved the protocol and the consent forms. Informed consent was obtained for all subjects in the COV-2069 and COVE trials. The ClinicalTrials.gov numbers were NCT04470472 and NCT04452318 for COVE and COV-2069, respectively.
