## [Peer Review File · Nature Communications]

Examining Protective effects of SARS-CoV-2 Neutralizing antibodies after vaccination or monoclonal antibody administrationReviewers' Comments:

Reviewer #1:

Remarks to the Author:

Positive feedback

It is very nice that the authors had access to very clean data from household exposure and protection by nAb trial. The analysis seems reasonable for most parts (see comments), and most details of the data and analyses are relatively well describes. The analysis of comparing protection efficacy by nAbs vs. mRNA vaccine is reasonable (although there are alternative ways of doing it). Overall, this is a statistically solid paper with good analyses.

Major concerns

Several mathematical details of the analysis require improvement to judge if the analysis is truly valid. For example, the model for predicted nAb titer is not described by equations (in lines 184-190), only by words. The details from supplement must be put here. Why was this approach chosen for analysis? Why not mixed effect modeling that is more common to describe PKPD of drugs for diverse groups of people? This must be made better. Also, I do not see any alternative analyses/assumptions for the model. E.g., why was the specific choice for models (suppl. lines 67-70)? Authors should explore whether any alternative ways of choosing the parameter distribution results in different answers.

Following up on the math. Is the approach to model censored data appropriate? This is a big field in statistics and some arguments are needed to justify better the chosen approach with citing appropriate literature. E.g., one example of how censored data could be used in likelihood methods (PMID: 17489970).

Why models (line 102, 108) are good models? What would be alternatives? One question I have is how to relate Ab titer to protection. Authors for some reason chose \log_{10} Ab as protection predictor. That seems to be inconsistent with physics/biology where protection should be linearly-dependent on Ab levels. Other models are also possible (e.g., see PMID: 32453765 for examples of alternative models a rigorous modeling study tries). Please investigate.

The way of how errors in estimated parameters were estimated was not well described. Authors cite "bootstrap" - line suppl 118 or "parametric bootstrap" line 203) but not very clear what "parameters" are bootstrapped and how. This must be better described. Providing codes for the analysis would be even better (along with the data!)

I understand that IU/ml is the agreed upon units for Abs against COVID19. Some discussion is needed of how this is important, e.g., relate that to IC50 that is more typically calculated and used in analyses. How wrong can IC50 be for the analyses that are done here? (i.e., could you use IC50 and then Ab titer/IC50 to evaluate efficacy of two treatments)? Can you also translate IU50/ml to ug/ml for these specific Abs? (e.g., line 180-81).

To test equivalency of two protection parameters you could fit the data from all trials together assuming either one efficacy or different efficacies and compare using likelihood ratio test. This would be a more powerful approach than bootstrap. And it will give you true p value. Bootstrap cannot really give p values (line 249).

Authors state in discussion that having nAb titers for all individuals prior to exposure would be useful but not practical. Can they perform some power analyses to indicate how that could impact their conclusions?

Did your models fit the data well? Some evaluation of the quality of the model fit to data (and alternative models tested) must be presented.

Minor concerns

It is unfortunate that the data from 2 published trials and from COVE trial are not publicly available. This is unacceptable. The prescribed recipe by authors to share the data is also unacceptable. I could think of 2-4 groups that could do the same analysis if they had access to the data, and with different teams approaching the same PUBLISHED data we would have more solid results at the end, benefitting all humanity!

When mentioning data from COVE there should be a citation of where the data came from. If this is a new study, this must be stated.

In papers involving modeling, equations must be numbered to authors and reviewers can refer to them.

I think some of the mathematical details - e.g., models for nAb dynamics and protection should be in the main text, not supplement. As this forms the basis of the paper, readers should have it upfront without looking for supplement.

I understand the potential issues with patient privacy but given that trials were done prior to Delta, this is probably already not an issue.

Sup. Line 114-115. This fit by Poisson model should be explicitly described and shown.

Line suppl. 137 - study participants are weighted by ... - Why? Explicit justification is needed.

Why does the first author have Phd to the name and others don't. Is that because others don't have degrees? Seems like a bias here.

Line 166 - 95 pound individual. Is that a child? I can hardly imagine this for an adult. Please discuss in more detail.

Decay rate of Abs cannot be log₁₀ IU₅₀/ml per day. Decay rates are in per day units. The proper way to model these is not with log₁₀ but with natural log. Please consult the expert on modeling Ab dynamics.

Line 282-283 - do we really know that nAbs are the major protector against COVID19? Please give the citation that accurately quantifies this!

285-286 - the fact that nAbs correlate with protection does not mean much. What does not correlate with protection?

Figure 1 - typically it is better to have placebo and then treatment groups discussed.

Figure 2 - I could not read any axes, I could not understand different lines (B&W printer). This must be improved, and add legends to plants to explain things - Add A/B/C to different panels.

Figure 3- put legend to explain what you want to say with thesis figure, What are the groups (I could not tell from B&W print out)

Figure 4: Why is the coloring of CIs for PE is with different colors. Caption does not explain it.

Figure 5: you may want to add information on the kinetics of Ab decay - e.g., half-life per patient or by fitting decay models with mixed effect models.

Figure 6: add quantitative numbers to what low titer and high titer are in the figure.

Figures S1: can you quantify these dynamics? Otherwise, showing them seems not very helpful. What should I be looking for here?

Fig S2: using least squares is inappropriate here because error is in both x/y axes. Use Deming regression or similar.

Figure S3: show numbers of patients and test statistics as done in proper figures on clinical trials.

S4: I cannot read axes. Can you also show the data fitted? Same for Fig S5

Reviewer #2:

Remarks to the Author:

Thank you for inviting me to review this manuscript. The authors evaluated NAb correlates of protection in COV-2069 (Casivirimab+Indevimab prophylaxis) trial and the mRNA-1273 vaccine COVE trial. The results have significant practical implications for how new monoclonal antibody authorisation could be viewed moving forwards, given the discrepancy between use of neutralisation surrogates for emergency authorisation of booster vaccines doses, whereas new MABs require clinical efficacy endpoints are present. A particular strength is that the authors standardise their output to WHO IU50/mL, allowing comparison with other correlate of protection outputs.

For the MAB that was assessed, the authors conclude that predicted NAb titres at the time of exposure of 1000 IU50/mL was associated with protective efficacy of 92% (84-98%), with a PE of >80% achieved with a NAb titre of 398 IU50/mL (25-631), and a titre of 200 IU50/mL (5-501) associated with a 50% PE.

For mRNA-1273, a titre at the time of exposure to 100IU50/mL was associated with a VE of 93% (91-95%) and a titre of 1000 IU50/mL associated with 97% (95-98%) VE.

Figure 4 demonstrating the relationship between VE and PE at given NAb titres for vaccines and MAB respectively is especially interesting and important, and a particular strength of the approach to compare the two.

A major limitation is of course that in effect only efficacy against pre-delta/pre-omicron SARS-CoV-2 could be assessed, which the authors acknowledge.

I have minor comments and suggestions:

1. For the COV-2069 study, the authors state that the study period was Jul 2020 - Oct 2021, yet no delta infections were included. This is unusual given the duration when delta circulated globally, but I assume it is because the recruitment period was early in this window. Please clarify this in the text.
2. It isn't fully clear how the IU50/mL equivalents of MAB concentration were generated and in which samples (what source/how chosen). Please expand on this in the methods.

3. The issue of whether vaccine induced antibodies may have confounded the MAb concentration vs PE in COV-2069 needs some further clarification and discussion. Looking at Herman et al, vaccines were prohibited prior to randomisation but allowed during follow up (with advise to follow CDC recommendations to receive these >90 days following MAb). 34.8% received at least 1 dose during the follow up period, with a median time to first vaccine 108.5 and 109 days in the two groups. It looks like only 3 participants developed symptomatic infection AFTER vaccination (1 in MAb and 2 in placebo), which suggests that vaccine effects should not have had much effect. This is worth stating up front.

4. The authors mention the total group sizes for the COVE trial. If I am not mistaken, looking at Gilbert et al, the serum substudy included antibody data from 36 post-day 57 cases and 1005 non-cases. This should be stated if the data derived for analysis of predicted NAb at time of exposure was from this sub-cohort.

5. Please include the median FU duration of the studies in the text (I appreciate it is shown in the figure)

Reviewer #3:

Remarks to the Author:

In this manuscript, Follmann and colleagues correlate neutralizing serum titers after monoclonal antibody administration with protection from symptomatic COVID-19 observed in a clinical trial. Determining a quantifiable correlate of protection (i.e., neutralization titers) for monoclonal antibodies may aid the development of antibody-based strategies for COVID-19 prevention. They compare titers and protection after antibody infusion with those determined in a mRNA-1273 vaccine trial. Based on similar protection at lower neutralization titers in the vaccine trial, they conclude that vaccines provide protection that goes beyond antibody-mediated neutralization.

Serum neutralization for the analyses was not specifically measured longitudinally but modeled/predicted. For the monoclonal antibodies, this was based on previously determined pharmacokinetic data of antibody concentrations and a correlation of some datapoints with experimentally determined neutralization titers. For the vaccine study, neutralization titers were predicted based on a decay factor derived from longitudinal neutralization data previously determined in vaccinees using a comparable pseudovirus assay

Strengths of the manuscript include

- Both the monoclonal antibody trial and the vaccine trial were conducted prior to the emergence of the Delta and Omicron variants of SARS-CoV-2 (and not in an environment with high rates of Beta circulation). Therefore, the results of both trials are broadly comparable from the perspective of neutralizing antibody sensitivity of the exposing viral strains.
- Neutralization titers are provided as WHO International Standard Units against (ancestral) D614G pseudovirus and the (limited number of) experimentally obtained neutralization titers was determined using comparable pseudovirus assays.
- Many limitations are appropriately acknowledged in the manuscript.

Main limitations and suggestions:

- The very wide confidence intervals due to the small number of cases prevent drawing firm conclusions for neutralization titers correlating with monoclonal antibody-mediated protection below

approximately 300 IU50/ml. In addition, low neutralization titers for the vaccine study could not be assessed because of the limited follow-up period. The actual data on which the lower half of Figure 6 ("low titer") is based, is therefore quite limited.

- In the Supplementary Materials on the "Deconstruction analysis" to quantify extant nAb effects, the authors describe that they separate (1) circulating and mucosal antibodies and (2) all other aspects of vaccine-induced immunity. This is also partially reflected in the legend to the scheme shown in Figure 6 ("role of extant circulating and mucosal antibody in vaccine induced protection"). All analyses are, however, based on the assessment of neutralizing serum titers as no mucosal samples were analyzed. Whether mucosal neutralizing activity is comparable for recipients of monoclonal antibodies and vaccinees is not shown.

- In their description of nAb titers (Figure 5), the authors state that "it is clear that an anamnestic response tends to be invoked in those with lower titers at symptom onset". From looking at the Figure, it is not necessarily clear that individuals with lower titers show an increase in neutralization after breakthrough activity but individuals with higher titers do not. It would be helpful to provide a stratified analysis with individuals grouped into "high" and "low" neutralization titers. Whether the stated observations hold true for individuals with asymptomatic infection remains speculative (all studied individuals had symptomatic infections).

- All analyses and results apply to the pre-Omicron era (as briefly stated in the section on study limitations). Neutralization titers against Omicron after two mRNA vaccine doses are typically very low. After post-vaccination breakthrough infection, a strong increase in neutralizing serum titers against Omicron has repeatedly been demonstrated (recall response, including for breakthrough infections with a non-Omicron variant). Nevertheless, vaccine effectiveness against Omicron-mediated symptomatic COVID-19 after two mRNA vaccine doses is very low. Thus, symptomatic infection can frequently occur in vaccinated individuals with low neutralization titers despite a nAb recall response. This aspect should be highlighted more. This is especially true as non-nAb-based mechanisms of protection are likely not as affected by Omicron mutations, even more suggesting a role for high antibody titers in preventing symptomatic COVID-19.

- In the section on study limitations, the authors note that testing asymptomatic infection in the COV-2069 trial was participant-driven. To my understanding, asymptomatic infections were not included in this manuscript as only protection from symptomatic COVID-19 cases was analyzed. This statement should perhaps be removed to avoid confusion.

We respond to each reviewer comment in italics below. Line references included match the clean version of the revised manuscript.

REVIEWER COMMENTS

Reviewer #1 (Remarks to the Author):

Positive feedback

It is very nice that the authors had access to very clean data from household exposure and protection by nAb trial. The analysis seems reasonable for most parts (see comments), and most details of the data and analyses are relatively well described. The analysis of comparing protection efficacy by nAbs vs. mRNA vaccine is reasonable (although there are alternative ways of doing it). Overall, this is a statistically solid paper with good analyses.

Thank you!

Major concerns

1) Several mathematical details of the analysis require improvement to judge if the analysis is truly valid. For example, the model for predicted nAb titer is not described by equations (in lines 184-190), only by words. The details from supplement must be put here. Why was this approach chosen for analysis? Why not mixed effect modeling that is more common to describe PKPD of drugs for diverse groups of people? This must be made better. Also, I do not see any alternative analyses/assumptions for the model. E.g., why was the specific choice for models (suppl. lines 67-70)? Authors should explore whether any alternative ways of choosing the parameter distribution results in different answers.

The predicted nAb titer equation (lines 184-190) was based on 18 samples selected from COV-2069 samples that covered the range of follow-up and that had both concentration and neutralization titer measured. These data points are displayed in Figure 2B and show a clear linear relationship between log concentration and log ID50. We thus estimated a linear regression using least squares regression obtaining $Y = 1.80 + 1.15 X$ (Figure 2 Panel B), where X is the log₁₀ concentration of CAS+IMB and Y the log₁₀ ID50 neutralization titer. Given the striking linear relationship we did not consider other models. The rewritten section (lines 1626-176) is given below which also now describes the population PK modeling used to generate the concentration curves

“For COV-2069 trial participants, pharmacokinetic (PK) antibody concentration curves of casirivimab and imdevimab in serum over 8-months were estimated using population PK models developed for casirivimab and imdevimab from three clinical studies. The population PK models were two compartment models with linear elimination and first-order absorption following subcutaneous dosing (25) Concentrations of casirivimab and imdevimab at each time point were added to obtain concentrations of casirivimab and imdevimab combined in serum. Stochastic simulations with inter-individual random effects were used to predict casirivimab and imdevimab concentration by weight and sex for each day of follow-up. Figure 2A displays the concentration time profiles for 10 randomly selected

individuals. To transform antibody concentration to pseudo-virus neutralization titer, we conducted a separate experiment where 27 frozen serum samples from COV-2069 participants that spanned the course of follow-up were selected. Eighteen samples had both concentration measured and neutralization titer above the limit of detection. We estimated the relationship between the concentration of antibodies and the neutralization titer for these 18 samples as $\log_{10}(\text{ID}_{50}) = 1.80 + 1.15 \times \log_{10}(\text{concentration})$, Figure 2B. We used this equation to determine individualized neutralization titer decay curves (Figure 2C). “

Regarding the linear model on lines 67-70 we also fit a model that allowed a bend-point to accommodate biphasic decay but this had virtually the same fit as the simpler model with linear decay of \log_{10} titer. The Wald statistics p-value for testing a bend-point was 0.54.

Following up on the math. Is the approach to model censored data appropriate? This is a big field in statistics and some arguments are needed to justify better the chosen approach with citing appropriate literature. E.g., one example of how censored data could be used in likelihood methods (PMID: 17489970).

Censoring could apply both to the antibody data (censored at the limit of detection) or the time to COVID-19 (censored by the end of follow-up).

We use standard Cox regression models for the censored time to event data. Such models were used for the COVE analysis (Gilbert, Peter B., et al. "Immune correlates analysis of the mRNA-1273 COVID-19 vaccine efficacy clinical trial." *Science* 375.6576 (2022): 43-50) and many other COVID-19 vaccines trials. See supplemental materials, Vaccine induced antibody risk model section.

In the supplemental materials "Modelling antibody kinetics of mRNA-1273" section, three datapoints were below the limit of detection. We used a likelihood-based analysis and the model assumes that antibody decline continues log-linearly below the limit of detection and censoring is a feature of the assay, but not the underlying biological process. The model integrates over the possible values below the limit of detection.

The datasets utilized for the development of population PK models for casirivimab and imdevimab were comprised of approximately 11,000 samples each. For casirivimab and imdevimab, <1.78% and <1.56% of these samples were below the limit of quantitation (BLQ), and as such, the BLQ samples were excluded from analysis.

3) Why models (line 102, 108) are good models? What would be alternatives? One question I have is how to relate Ab titer to protection. Authors for some reason chose \log_{10} Ab as protection predictor. That seems to be inconsistent with physics/biology were protection should be linearly-dependent on Ab levels. Other models are also possible (e.g., see PMID: 32453765 for examples of alternative models a rigorous modeling study tries). Please investigate.

Thank you for the opportunity to clarify. Indeed, model evaluation was undertaken. We used likelihood ratio tests to consider different models for COV-2069 as given below.

$$h(d) = h_0(d) \exp\{Z[\beta_0]\}, \quad (\text{A})$$

$$h(d) = h_0(d) \exp\{Z[\beta_0 + \beta_1 \text{Ab}(d)]\}, \quad (\text{B})$$

$$h(d) = h_0(d) \{ Z \expit\{\beta_0 + \beta_1 Ab(d)\} \} \} \text{ (C)}$$

$$h(d) = h_0(d) \{ (1 - Z) + Z[\theta + (1 - \theta) \expit\{\beta_0 + \beta_1 Ab(d)\} \} \} \text{ (D)}$$

where d is days since infusion and $Ab(d)$ is the predicted log10 ID50 neutralization titer at day d . Model (A) has only an intercept term with no effect of antibody, (B) is the 'standard' model used in multiple COVID-19 vaccine trials with a linear term of $Ab(d)$ (see e.g. Gilbert, Peter B., et al. "A Covid-19 Milestone Attained—A Correlate of Protection for Vaccines." *New England Journal of Medicine* 387.24 (2022): 2203-2206, while (C) and (D) are 2 and 3 parameter logistic models. The likelihood ratio test p -values are (B)to(A) 0.011 (A)vs(C) 0.009 (A)vs(D)=0.006 and (C)vs(D) = 0.056. Importantly all models show significantly improved fit when $Ab(d)$ is included in the model thus demonstrating robustness of our conclusion that titer correlates with protection for this mAb.

MODEL	Hazard	-log likelihood
A) Intercept Only	$h_0(t) \exp\{Z[\beta_0]\}$,	664.25
B) Log-Linear	$h_0(d) \exp\{Z[\beta_0 + \beta_1 Ab(d)]\}$,	660.82
C) 2 parameter logistic (2PL)	$h_0(d) \{ Z \expit\{\beta_0 + \beta_1 Ab(d)\} \}$	661.01
D) 3 parameter logistic (3PL)	$h_0(d) \{ (1 - Z) + Z[\theta + (1 - \theta) \expit\{\beta_0 + \beta_1 Ab(d)\} \} \}$	659.19

We chose the 3PL based on these tests and on biological plausibility---PE=0 with no antibodies, but it also asymptotes to $(1 - \theta)$ as $Ab(d)$ goes to infinity. For completeness, we also report the log-linear model (B) in the supplementary materials in Figure S6 to contrast with the analogous vaccine VE curve. Use of the log-linear model results in somewhat stronger conclusions compared to the 3PL.

We added the following text to the paper (lines 211-212) and cite the Gilbert study referenced above: "Log linear VE curves were used in a prior analysis of COVE and multiple vaccine trials thus allowing direct comparisons with other studies.(34)"

Your second question concerns choice of log10 Ab. This is the standard readout for neutralization antibodies in the USG sponsored COVID-19 vaccine studies, so its use here facilitates comparison. For thoroughness we tried to fit model (B) using ID50 titer as the covariate instead of $Ab = \log_{10}(ID50)$, but this model did not converge.

Note that for these log-linear hazard models, risk is proportional to $\exp()$ of log10-antibody ID50 titer. Thus, in model B, risk is proportional to a power of ID50 titer. To see this, assume for simplicity that time to infection follows an exponential distribution. Define $Ab = \log_{10}(x)$ where x is the predicted ID50 neutralization titer. Thus, the hazard for COVID-19 for a mAb participant is

$$h_0 \exp\{\beta_0 + \beta_1 Ab\} = h_0 \exp\{\beta_0\} \exp\{\beta_1 Ab\} = h_0 \exp\{\beta_0\} \exp\{\beta_1 \log_{10}(X)\} =$$

$$h_0 \exp\{\beta_0\} \exp\{\beta_1 \ln(X) \log_{10}(e)\} = h_0 \exp\{\beta_0\} X^{\beta_1 \log_{10}(e)}$$

Note β_1 is a parameter that can assume any value on the real line and thus can equal $1/\log_{10}(e)$ in which case risk is directly proportional to titer. Other values of β_1 , are possible and allow for a richer class of models.

4) The way of how errors in estimated parameters were estimated was not well described. Authors cite "bootstrap" - line 118 or "parametric bootstrap" line 203) but not very clear what "parameters" are bootstrapped and how. This must be better described. Providing codes for the analysis would be even better (along with the data!)

Thanks for bringing this up, indeed it was an involved process and not clearly described. A new description is given in the supplementary materials Monoclonal antibody risk model section, which we provide below.

We obtain 95% pointwise confidence intervals for the parameters in (S2) <i.e. model (D) above> via the bootstrap and propagate uncertainty about the relationship between concentration and titer by a random draw from the bivariate distribution of the slope and intercept as estimated from the 18 paired samples. Details follow. We first estimated the parameters from the below equation

$$Y_i = \eta_0 + \eta_1 X_i + e_i$$

using Y_i, X_i $i=1, \dots, 18$ the paired samples of \log_{10} (ID50), \log_{10} concentration of antibody. Estimation was done by ordinary least squares. Denote the estimated parameters as $\hat{\eta}_0, \hat{\eta}_1$ and the estimated covariance matrix of $\hat{\eta}_0, \hat{\eta}_1$ by \hat{C} . For a single bootstrap sample of the 1630 individuals in the COV-2069 trial we first sampled η_0^b, η_1^b from a bivariate normal distribution with mean $\hat{\eta}_0, \hat{\eta}_1$ and covariance \hat{C} . From this η_0^b, η_1^b we generated individualized \log_{10} (ID50) decay curves by creating, for each day and each person the predicted \log_{10} ID50 titer, y_{jd} , according to the equation

$$y_{jd} = \eta_0^b + \eta_1^b x_{jd}$$

where x_{jd} was the antibody concentration for person $i=1, \dots, 1630$ on day $d=1, \dots, 240$ post injection. We then sampled the 1630 participants in the analysis set with replacement. Using these 1630 sampled participants we estimated the parameters in the hazard function (S2). We did this 10,000 times resulting in 10,000 estimates of θ, β_0, β_1 .

COVE is similar and we write the following in the Vaccine induced antibody risk model section.

We obtain 95% pointwise confidence intervals via a two-step procedure similar to COV-2069. For each bootstrap sample we sample the COVE participants with replacement. We propagate uncertainty about the rate of neutralizing antibody decay estimated from the previously described antibody model by using a random draw from the posterior distribution of the rate of decay in each bootstrap iteration to predict neutralization titers at each day post day 57. Following Gilbert, et al.,(8) the resampling step of COVE participants was stratified by groupings of risk-demographic strata and randomization to the immunogenicity subcohort.

5) I understand that IU/ml is the agreed upon units for Abs against COVID19. Some discussion is needed of how this is important, e.g., relate that to IC50 that is more typically calculated and used in analyses. How wrong can IC50 be for the analyses that are done here? (i.e., could you use IC50 and then Ab titer/IC50 to evaluate efficacy of two treatments)? Can you also translate IU50/ml to ug/ml for these specific Abs? (e.g., line 180-81).

*The main attraction of IU/ml is to facilitate comparison with multiple COVID-19 vaccine trials (see e.g. Gilbert, Peter B., et al. "A Covid-19 Milestone Attained—A Correlate of Protection for Vaccines." *New England Journal of Medicine* 387.24 (2022): 2203-2206.)*

To help with translation, we have added a sentence to line 192-195 which now reads.

“Neutralization titers are reported in international units as $IU_{50}/ml = ID_{50} \times 0.242$ where ID_{50} is the reciprocal ID_{50} dilution titer (see Gilbert et al.,(8) and supplementary materials). Thus, a titer in IU_{50}/ml can be converted to ug/ml by dividing by 0.242, e.g. 1000 IU_{50}/ml corresponds to a titer of 4132.2 ug/ml .”

6) To test equivalency of two protection parameters you could fit the data from all trials together assuming either one efficacy or different efficacies and compare using likelihood ratio test. This would be a more powerful approach than bootstrap. And it will give you true p value. Bootstrap cannot really give p values (line 249).

We agree likelihood ratio tests are more powerful and could work if we used the same hazard function for both trials, but for COV-2069 we use the 3-parameter logistic curve and for COVE the log-linear model which are non-nested models precluding a likelihood ratio test. Further, the models use different time indices; calendar time for COVE and time since randomization for COV-2069. This led us to use a bootstrap test, where we calculate the difference $\widehat{VE}(1.6)^b - \widehat{PE}(1.6)^b$ for $b=1, \dots, 10,000$ bootstrap samples and record the twice proportion of times this difference is less than 0 (see Efron & Tibshirani 1993 “An Introduction to the Bootstrap” Chapter 16 and Rousselet, et al, 2021, Advances in Methods and Practices in Psychological Science, “The Percentile Bootstrap: A Primer with Step-by-Step Instructions in R”, Vol 4, No.1, 1-10).

To approximate the more powerful likelihood ratio test for nested models, consider Figure S6 which displays the VE and PE curves based on the common log-linear model

$$h(d) = h_0(d) \exp\{Z[\beta_0 + \beta_1 Ab(d)]\},$$

applied to both trials (with some baseline covariates for COVE). Due to the different time index, we tested whether β_0, β_1 were the same for the two trials using a generalized Wald test. The p-value for this test was 0.001, which we now report in the text to demonstrate robustness of our results to different models. Lines 219-221 of the paper now state, “A bootstrap approach was used to test equality of the PE and VE curves at specific titers and a generalized Wald test used to test overall equality of the log-linear VE and PE curves.” And Lines 268-269 “A generalized Wald test of equality of the log-linear PE and VE curves in Figure S6 rejects at $p < 0.001$.”

7) Authors state in discussion that having nAb titers for all individuals prior to exposure would be useful but not practical. Can they perform some power analyses to indicate how that could impact their conclusions?

Use of actual titer would presumably sharpen our results because actual titer would be a more precise measure of the biology on any given day. Necessity required this simpler approach.

To get a handle on the effect of using predicted titer instead of actual titer, one can view our approach as a regression calibration method to the problem of a covariate measured with error. For simplicity consider a linear regression model

$$E(Y) = B_0 + B_1 x$$

where x is the true/actual value of the covariate, but we use a covariate \hat{X} which is the expected value of x based on a model (i.e. that day’s predicted assay readout). Regression calibration results in an attenuated estimate of B_1 . The degree of attenuation depends on the variation in x in the study compared to the variance of the prediction error ($\hat{X} - x$). For COVE, we conservatively approximate $\text{var}(x)$

by the sample $\text{var}(\hat{X})$ throughout follow-up and approximate the variance of prediction error $(\hat{X}-x)$ from the error variance estimate from the Doria-Rose data. These estimates are $\text{var}(\hat{X}) = 1.10$ and $\text{var}(\hat{X}-x) = 0.02$ and thus $\text{var}(x)/(\text{var}(x) + \text{var}(e))$ is approximated by 0.98. This is a setting where regression calibration should work quite well with little attenuation. (see e.g., Follmann, D. A., Hunsberger, S. A., & Albert, P. S. (1999). Repeated probit regression when covariates are measured with error. *Biometrics*, 55(2), 403-409.)

While these calculations are more difficult for COV-2069 we anticipate $\text{var}(x)/\{\text{var}(x) + \text{var}(e)\}$ would be even larger due to the greater variability in titer over follow-up for the mAb (see Figure 3).

8) Did your models fit the data well? Some evaluation of the quality of the model fit to data (and alternative models tested) must be presented.

To assess goodness of fit we compared the Kaplan-Meier curves of cumulative incidence with the model-based cumulative incidence, see below for the COVE and COV-2069 studies, respectively. A feature of these data is that the placebo arm is fit quite well because the vast majority of cases are from the placebo arms in both trials. We see that the model based cumulative incidence curve for the vaccine arm lies within the confidence band for the Kaplan-Meier curve and nearly so for the mAb arm which is more variable due to having few events in that arm.

COVE VACCINE TRIAL

Cumulative incidence estimates - placebo arm

Model (dashed red line), Kaplan-Meier (solid black) + 95% CI (grey band)

Cumulative incidence estimates - Vaccine arm

Model (dashed red), Kaplan-Meier (solid black) + 95% CI (grey band)

COV-2069 MONOCLONAL ANTIBODY TRIAL

Cumulative incidence estimates - placebo arm

Model (dashed red line), Kaplan-Meier (solid black) + 95% CI (grey band)

Cumulative incidence estimates - mAb arm

Model w/titre (dashed red), Model w/treatment alone (dotted blue), Kaplan-Meier (solid black)

Another approach to assess goodness of fit for proportional hazards models is to see if the regression parameters (slopes) differ for early versus late follow-up. We thus interacted the coefficients for arm and arm*log10(titer) for COVE with an indicator of early versus late follow-up. Neither interaction term was significant for COVE. For COV-2069 the model could not be fit due to too few events.

We include the above information in the supplemental materials, section Model Goodness-of-Fit

In prior comments we evaluated alternative models for COV-2069. Below we fit some alternative models for COVE along with the log-likelihoods for each.

Model	Model Equation	Degrees of freedom	-log likelihood
Intercept Only	$h_0(d) \exp\{Z[\beta_1]\}$,	4	6564.5
Log-Linear	$h_0(d) \exp\{Z[\beta_0 + \beta_1 Ab(d)]\}$,	5	6560.4
Quadratic	$h_0(d) \exp\{Z[\beta_0 + \beta_1 Ab(d) + \beta_2 Ab(d)^2]\}$,	6	6557.7
Spline	$h_0(d) \exp\{Z[\beta_0 + f\{Ab(d)\}]\}$, where $f()$ is a natural cubic spline with three degrees of freedom and interior knots set at the first and second tertiles of the predicted log-titre distribution (0.48, and 1.24).	7	6557.8

All models with $Ab(d)$ in them are preferred to the intercept model (all p -values <0.01), robustly showing that there is a relationship between titer and protection. The linear model is preferred to the intercept model ($p<0.01$) and the spline model does not show significant improvement over the linear model ($p=0.07$). While the quadratic offers a somewhat improved fit to the linear model ($p=0.02$), its estimates of VE are similar to the main model (92% versus 93% at a titer of 100 and 97% vs 99% at a titer of 1000). In a sensitivity analyses we performed a bootstrap test of equality of VE and PE at a log10 titer of 1.6 comparing the 3PL PE model for mAbs with the quadratic model VE for COVE. The p -value for this test was 0.04, similar to the 0.03 we obtained comparing the 3PL PE model to the linear VE model for COVE, demonstrating robustness of this result to model specification.

We prefer the main model because this was our planned analysis for the COVE trial and it facilitates comparison with the results of the other USG sponsored trials which all used this model (see e.g., Gilbert, Peter B., et al. "A Covid-19 Milestone Attained—A Correlate of Protection for Vaccines." *New England Journal of Medicine* 387.24 (2022): 2203-2206.)

Minor concerns

1) It is unfortunate that the data from 2 published trials and from COVE trial are not publicly available. This is unacceptable. The prescribed recipe by authors to share the data is also unacceptable. I could think of 2-4 groups that could do the same analysis if they had access to the data, and with different teams approaching the same PUBLISHED data we would have more solid results at the end, benefitting all humanity!

We understand this concern and added the following specific data availability statements to the paper based on the study sponsors.

For COV-2069

Data availability: Qualified researchers can request access to study documents (including the clinical study report, study protocol with any amendments, blank case report form, and statistical analysis plan) that support the methods and findings reported in this manuscript. Individual anonymised participant data will be considered for sharing once the product and indication has been approved by major health authorities (eg, US Food and Drug Administration, European Medicines Agency, Pharmaceuticals and Medical Devices Agency, and so on), if there is legal authority to share the data and there is not a reasonable likelihood of participant re-identification. Requests should be submitted to <https://vivli.org/>.

For COVE

As the trial is ongoing, access to participant-level data and supporting clinical documents with qualified external researchers may be available upon request and is subject to review once the trial is complete. Such requests can be made to Moderna Inc., 200 Technology Square, Cambridge, MA 02139, USA. A materials transfer and/or data access agreement with the sponsor will be required for accessing of shared data. All other relevant data are presented in the paper. The protocol is available in the Supplementary Information: Clintrials.gov. NCT04470427.

2) When mentioning data from COVE there should be a citation of where the data came from. If this is a new study, this must be stated.

In lines 138-138 we clarify that the analysis of Gilbert et al., is through the end of the blinded phase and state we use this dataset on line 142. More details about the dataset are included in the supplemental materials Vaccine induced antibody risk model section.

We also use data from Follmann, Janes et al., and state this on lines 149 of the paper.

3) In papers involving modeling, equations must be numbered to authors and reviewers can refer to them.

We now include equation numbers in both the paper and the supplemental materials.

4) I think some of the mathematical details - e.g., models for nAb dynamics and protection should be in the main text, not supplement. As this forms the basis of the paper, readers should have it upfront without looking for supplement.

We now provide the equation linking the mAb concentration to neutralization titer as well as the PE and VE hazard models, with equations numbers, in the main text and state on lines 209-218:

“For COVE, a log-linear curve was used

$$VE(Ab) = 1 - \exp(\beta_0 + \beta_1 Ab) \}, (1)$$

Log linear VE curves were used in a prior analysis of COVE and multiple vaccine trials thus allowing direct comparisons with other studies.(34) For COV-2069 a three-parameter logistic curve was used

$$PE(Ab) = 1 - \{ \theta + (1 - \theta) \expit(\beta_0 + \beta_1 Ab) \}, (2)$$

where Ab is $\log_{10}(ID50)$ and $\expit(a) = \exp(a)/(1 + \exp(a))$. This curve allows for flexible modeling of a wide range of PEs. The maximal protective efficacy is given by $(1 - \theta)$, the ratio $-\beta_0/\beta_1$ determines the level of Ab where the maximal protective efficacy of $(1 - \theta)$ is halved, and as $ID50$ goes to 0, PE goes to zero. We also estimated a log-linear curve for PE as a sensitivity analysis.”

5) I understand the potential issues with patient privacy but given that trials were done prior to Delta, this is probably already not an issue.

See above for the company policies regarding data sharing.

6) Sup. Line 114-115. This fit by Poisson model should be explicitly described and shown.

We describe the model and report bootstrap confidence intervals for the 50% PE and saturation parameters in lines 257-258. The 50% PE is similar to but more meaningful than the halving parameter. We added the following to the supplemental materials, Monoclonal antibody risk model section:

“We first estimated the parameters from the below equation

$$Y_i = \eta_0 + \eta_1 X_i + e_i \quad (S5)$$

using Y_i, X_i $i=1, \dots, 18$ the paired samples of $\log_{10}(ID50)$, \log_{10} concentration of antibody. Estimation was done by

ordinary least squares. Denote the estimated parameters as $\widehat{\eta}_0, \widehat{\eta}_1$ and the estimated covariance matrix of $\widehat{\eta}_0, \widehat{\eta}_1$ by \widehat{C} .

For a single bootstrap sample of the 1630 individuals in the COV-2069 trial we first sampled η_0^b, η_1^b from a bivariate normal distribution with mean $\widehat{\eta}_0, \widehat{\eta}_1$ and covariance \widehat{C} .

From this η_0^b, η_1^b we generated individualized $\log_{10}(ID50)$ decay curves by creating, for each day and each person the predicted \log_{10} ID50 titer, y_{id} , according to the equation

$$y_{id} = \eta_0^b + \eta_1^b x_{id} \quad (S6)$$

where x_{id} was the antibody concentration for person $i=1, \dots, 1630$ on day $d=1, \dots, 240$ post injection. We then sampled the 1630 participants in the analysis set with replacement. Using these 1630 sampled participants we estimated the parameters in the hazard function (MA). We did this 10,000 times resulting in 10,000 estimates of θ, β_0, β_1 . We calculated percentile bootstrap confidence intervals for different functions of θ, β_0, β_1 by determining the .025 and .975 percentiles of the 10,000 estimates. For example $PE^b(Ab)$, $b=1, \dots, 10,000$ where $PE^b(Ab)$ is equation (S2) with the parameters replaced by their estimated values using the b th bootstrapped data set.”

7) Line suppl. 137 - study participants are weighted by ... - Why? Explicit justification is needed.

We now introduce the details of the weighted VE estimation in the supplemental materials Vaccine induced antibody risk model section:

“Instead of assessing Day 57 antibody in all 14,202 vaccine arm participants Gilbert et al., (8) used a case-cohort design which was comprised of a stratified random sample of 1010 participants who comprised the immunogenicity sub-cohort, plus 36 disease cases (5 of which were in the set of 1010).”

More details are given after that, but in essence, each sampled person is weighted so that the weighted distribution of patient characteristics matches the distribution of characteristics of the 14,202.

8) Why does the first author have Phd to the name and others don't. Is that because others don't have degrees? Seems like a bias here.

Thanks for catching this, this was an inadvertent error on our part and has been corrected.

9) Line 166 - 95 pound individual. Is that a child? I can hardly imagine this for an adult. Please discuss in more detail.

We looked into this and found that it was an adult woman with small stature. Note that for a 4' 10" women a healthy weight ranges form 91 to 115 pounds. See <https://www.nhlbi.nih.gov/sites/default/files/media/docs/Session6-BMICChart-508.pdf>. We deleted this text to avoid confusion.

10) Decay rate of Abs cannot be log10 IU50/ml per day. Decay rates are in per day units. The proper way to model these is not with log10 but with natural log. Please consult the expert on modeling Ab dynamics.

*Thank you for your comment. We converted change per day in log10 units to natural logs to determine the half-life. Specifically, for the standard exponential decay formula $Y(t) = Y(0)\exp(-\lambda t)$, we determined $\lambda = \ln(10) B$ where B was the change per day in log10 units. This allowed us to determine the half-life for monoclonal Ab decay as $\log(2)/(\log(10)*0.012) = 26$ days and for vaccine Ab decay as $\log(2)/(\log(10)*0.0043) = 70$ days. We've rewritten this section so that we only report the half-lives in the results Neutralization Titer Kinetics section. We provide the above formula in the supplementary materials.*

“For participants in the monoclonal antibody COV-2069 trial, the median half-life of the neutralization titer was 26 days. For vaccinated individuals in the COVE trial, the estimated half-life was 70 days.”

11) Line 282-283 - do we really know that nAbs are the major protector against COVID19? Please give the citation that accurately quantifies this!

Thank you for pointing this out. There is not a citation that we know of that qualifies this statement, therefore it was deleted.

12) 285-286 - the fact that nAbs correlate with protection does not mean much. What does not correlate with protection?

The correlation of nAbs with protection was a central goal of the USG supported Operation Warp Speed vaccine trials. The establishment of this correlation has allowed the use of neutralization titer to predict efficacy for variants based on titers against variant pseudo-virus neutralization assays and has allowed

licensure of variant vaccines by regulatory agencies without running large Phase III clinical trials (see e.g., Gilbert, Peter B., et al. "A Covid-19 Milestone Attained—A Correlate of Protection for Vaccines." New England Journal of Medicine 387.24 (2022): 2203-2206. While other mechanisms likely contribute to protection, e.g., T cells, non-neutralizing functions of antibodies, memory B cells, etc., further data is needed to definitively quantitate the contribution. Studies to quantify the role of T cells and other mechanisms vis-à-vis antibodies are ongoing and of great interest.

13) Figure 1 - typically it is better to have placebo and then treatment groups discussed.

Thank you, the order has been reversed in figure 1.

14) Figure 2 - I could not read any axes, I could not understand different lines (B&W printer). This must be improved, and add legends to panels to explain things - Add A/B/C to different panels.

Thank you, we have modified figure 2 as suggested.

15) Figure 3- put legend to explain what you want to say with this figure, What are the groups (I could not tell from B&W print out)

Thanks, updated the legend for clarification.

16) Figure 4: Why is the coloring of CIs for PE is with different colors. Caption does not explain it.

This is meant to evoke that there is more uncertainty for titers below $10^{2.5}$. We now explain this in the caption.

17) Figure 5: you may want to add information on the kinetics of Ab decay - e.g., half-life per patient or by fitting decay models with mixed effect models.

Thank you, we now report in our results section, lines 285-287:

“Based on random effects modeling, the half-lives of antibody abundance from 28 days post 2nd dose to the onset of symptoms was estimated as 58 days for binding antibody and 62 days for neutralization antibodies.”

18) Figure 6: add quantitative numbers to what low titer and high titer are in the figure.

Low and high are not precise numbers but cover a range. We convey the idea of a range in the caption and describe higher as e.g., 1000 IU₅₀/ml and lower as e.g., < 100 IU₅₀/ml.

19) Figures S1: can you quantify these dynamics? Otherwise, showing them seems not very helpful. What should I be looking for here?

This is meant to show the data. We report the parameter estimates of the slope of decay, β_1 , and its confidence interval in the text of the supplementary materials.

20) Fig S2: using least squares is inappropriate here because error is in both x/y axes. Use Deming regression or similar.

Thank you, we now using Deming regression. We updated this in figure S2 and in the text of the paper (line 206).

21) Figure S3: show numbers of patients and tests statistics as done in proper figures on clinical trials.

We now provide the number at risk and report the log-rank p-value.

22) Figure S4: I cannot read axes. Can you also show the data fitted? Same for Fig S5

The labels and axes font sizes have been increased. The x-axis shows the predicted log₁₀ neutralization at disease onset for both the mAb and placebo arms and this has also been increased in size.

Reviewer #2 (Remarks to the Author):

Thank you for inviting me to review this manuscript. The authors evaluated NAb correlates of protection in COV-2069 (Casivirimab+Indevimab prophylaxis) trial and the mRNA-1273 vaccine COVE trial. The results have significant practical implications for how new monoclonal antibody authorisation could be viewed moving forwards, given the discrepancy between use of neutralisation surrogates for emergency authorisation of booster vaccines doses, whereas new MAbs require clinical efficacy endpoints are present. A particular strength is that the authors standardise their output to WHO IU50/mL, allowing comparison with other correlate of protection outputs.

Thank you!

For the MAb that was assessed, the authors conclude that predicted NAb titres at the time of exposure of 1000 IU50/mL was associated with protective efficacy of 92% (84-98%), with a PE of >80% achieved with a NAb titre of 398 IU50/mL (25-631), and a titre of 200 IU50/mL (5-501) associated with a 50% PE.

For mRNA-1273, a titre at the time of exposure to 100IU50/mL was associated with a VE of 93% (91-95%) and a titre of 1000 IU50/mL associated with 97% (95-98%) VE.

Figure 4 demonstrating the relationship between VE and PE at given NAb titres for vaccines and MAB respectively is especially interesting and important, and a particular strength of the approach to compare the two.

A major limitation is of course that in effect only efficacy against pre-delta/pre-omicron SARS-CoV-2 could be assessed, which the authors acknowledge.

I have minor comments and suggestions:

1. For the COV-2069 study, the authors state that the study period was Jul 2020 - Oct 2021, yet no delta infections were included. This is unusual given the duration when delta circulated globally, but I assume it is because the recruitment period was early in this window. Please clarify this in the text.

Thank you. We made an error in this statement and have changed it to read (lines 110-111): "The study was conducted prior to the emergence of Omicron-lineage VOC."

2. It isn't fully clear how the IU50/mL equivalents of MAb concentration were generated and in which samples (what source/how chosen). Please expand on this in the methods.

Twenty-seven samples were selected from COV-2069 to cover the course of follow-up. Criteria included not being vaccinated or seropositive, no evidence of prior RT-qPCR positive test, with a pristine sample and available backup samples. 18 of the samples had neutralization titers above the limit of detection with concentration data and were used to estimate the linear relationship between concentration and neutralization titer. We now write in the Neutralizing Antibody Data section (lines 170-178):

“To transform antibody concentration to pseudo-virus neutralization titer, we conducted a separate experiment where 27 frozen serum samples from COV-2069 participants that spanned the course of follow-up were selected. Eighteen samples had concentration measured and neutralization titer above the limit of detection. We estimated the relationship between the concentration of antibodies and the neutralization titer for these 18 samples was estimated as $\log_{10}(ID_{50}) = 1.80 + 1.15 \times \log_{10}(\text{concentration})$, Figure 2B.”

3. The issue of whether vaccine induced antibodies may have confounded the MAb concentration vs PE in COV-2069 needs some further clarification and discussion. Looking at Herman et al, vaccines were prohibited prior to randomisation but allowed during follow up (with advise to follow CDC recommendations to receive these >90 days following MAb). 34.8% received at least 1 dose during the follow up period, with a median time to first vaccine 108.5 and 109 days in the two groups. It looks like only 3 participants developed symptomatic infection AFTER vaccination (1 in MAb and 2 in placebo), which suggests that vaccine effects should not have had much effect. This is worth stating up front.

Thank you for this important point, we now state in the methods section (lines 117-120): “While vaccines were prohibited prior to randomization, 35% of participants reported receiving at least one dose during the follow-up period with a median of 109 days to first dose and were included in our modified intent-to-treat (mITT) analysis set. Three participants (two on placebo) developed symptomatic infection after vaccination.”

We also mention this ‘lack of compliance’ as a limitation in the discussion which might weaken estimated relationships. We state on lines 371-372, “In the COV-2069 trial some vaccination occurred which might weaken our results.”

4. The authors mention the total group sizes for the COVE trial. If I am not mistaken, looking at Gilbert et al, the serum substudy included antibody data from 36 post-day 57 cases and 1005 non-cases. This should be stated if the data derived for analysis of predicted NAb at time of exposure was from this sub-cohort.

In lines 142-144 we state:

Here, we use the same data as Gilbert et al.,(8) but correlate the incidence of COVID-19 events with predicted neutralization titers throughout follow-up, rather than at day 57, thus analyzing titer as an exposure proximal correlate.

We now introduce the details of the VE estimate with the following in the Supplementary Materials.

“Instead of assessing Day 57 antibody in all 14,202 vaccine arm participants Gilbert et al., (8) used a case-cohort design which was comprised of a stratified random sample of 1010 participants who comprised the immunogenicity sub-cohort, plus 36 disease cases (5 of which were in the set of 1010).”

5. Please include the median FU duration of the studies in the text (I appreciate it is shown in the figure)

*We now include, for COVE, “and the median follow-up was 157 days” on line 136.
We now state for COV-2069 on line 114, “The median follow-up was 225 days.”*

Reviewer #3 (Remarks to the Author):

In this manuscript, Follmann and colleagues correlate neutralizing serum titers after monoclonal antibody administration with protection from symptomatic COVID-19 observed in a clinical trial. Determining a quantifiable correlate of protection (i.e., neutralization titers) for monoclonal antibodies may aid the development of antibody-based strategies for COVID-19 prevention. They compare titers and protection after antibody infusion with those determined in a mRNA-1273 vaccine trial. Based on similar protection at lower neutralization titers in the vaccine trial, they conclude that vaccines provide protection that goes beyond antibody-mediated neutralization.

Serum neutralization for the analyses was not specifically measured longitudinally but modeled/predicted. For the monoclonal antibodies, this was based on previously determined pharmacokinetic data of antibody concentrations and a correlation of some datapoints with experimentally determined neutralization titers. For the vaccine study, neutralization titers were predicted based on a decay factor derived from longitudinal neutralization data previously determined in vaccinees using a comparable pseudovirus assay

Strengths of the manuscript include

Both the monoclonal antibody trial and the vaccine trial were conducted prior to the emergence of the Delta and Omicron variants of SARS-CoV-2 (and not in an environment with high rates of Beta circulation). Therefore, the results of both trials are broadly comparable from the perspective of neutralizing antibody sensitivity of the exposing viral strains.

Neutralization titers are provided as WHO International Standard Units against (ancestral) D614G pseudovirus and the (limited number of) experimentally obtained neutralization titers was determined using comparable pseudovirus assays.

Many limitations are appropriately acknowledged in the manuscript.

Thank you!

Main limitations and suggestions:

1) The very wide confidence intervals due to the small number of cases prevent drawing firm conclusions for neutralization titers correlating with monoclonal antibody-mediated protection below approximately 300 IU₅₀/ml. In addition, low neutralization titers for the vaccine study could not be assessed because of the limited follow-up period. The actual data on which the lower half of Figure 6 (“low titer”) is based, is therefore quite limited.

We agree that firm conclusions are hard to draw based solely on the VE vs PE comparison due to the small number of cases. The VE vs PE evidence is 1) a test that the curves do differ at the lowest titer of common follow-up ($p < 0.05$), 2) supplemental Figure S6 which fits PE and VE curves using the same model which shows wider separation at lower titers ($p < 0.001$), and 3) a similar looking though poorly estimated PE curve for asymptomatic infection, Figure S5.

We augmented the PE vs VE evidence with a test showing that vaccine COVID-19 cases tend to have an anamnestic response for those with low titers at disease presentation but not for those with high titers. We feel this result also supports the bottom part of Figure 6.

We edited and moved Figure 6 to the discussion to make clearer that this is an interpretation of the evidence (discussion, lines 323-325).

2) - In the Supplementary Materials on the “Deconstruction analysis” to quantify extant nAb effects, the authors describe that they separate (1) circulating and mucosal antibodies and (2) all other aspects of vaccine-induced immunity. This is also partially reflected in the legend to the scheme shown in Figure 6 (“role of extant circulating and mucosal antibody in vaccine induced protection”). All analyses are, however, based on the assessment of neutralizing serum titers as no mucosal samples were analyzed. Whether mucosal neutralizing activity is comparable for recipients of monoclonal antibodies and vaccinees is not shown.

We wanted to make clear that extant antibody was not entirely restricted to antibodies in the blood, even though we measured antibodies in the blood. It is true that we don't know the extent of penetration to the mucosa, but feel that the new phrasing in the figure 6 legend ‘circulating and possibly mucosal antibodies’ is more accurate than ‘circulating antibodies,’ which would imply that any mucosal antibodies belong in ‘bucket’ (2)-all other aspects of vaccine induced immunity. But there is evidence that circulating antibodies do migrate to the mucosa, see below for vaccines and mAbs.

Declercq J, Tobback E, Vanhee S, De Ruyck N, Gerlo S, Gevaert P, Vandekerckhove L. COVID-19 vaccination with BNT162b2 and ChAdOx1 vaccines has the potential to induce nasal neutralizing antibodies. Allergy. 2022 Jan;77(1):304-307. doi: 10.1111/all.15101. Epub 2021 Sep 29. PMID: 34543444; PMCID: PMC8653144.

Loo, Yueh-Ming, et al. "The SARS-CoV-2 monoclonal antibody combination, AZD7442, is protective in nonhuman primates and has an extended half-life in humans." Science translational medicine 14.635 (2022): eabl8124.

3) In their description of nAb titers (Figure 5), the authors state that “it is clear that an anamnestic response tends to be invoked in those with lower titers at symptom onset”. From at looking at the Figure, it is not necessarily that “clear” that individuals with lower titers show an increase in neutralization after breakthrough activity but individuals with higher titers do not. It would be helpful to provide a stratified analysis with individuals grouped into “high” and “low” neutralization titers. Whether the stated observations hold true for individuals with asymptomatic infection remains speculative (all studied individuals had symptomatic infections).

Thanks, for the opportunity to clarify. We now write in our results In mRNA-1273 Vaccine Disease Cases, nAB Titers Rise In those with Lower Titers section, lines 287-296:

“We identified those who had antibody measurements at both the onset of symptoms (Day 0) and 28 days later and calculated the median antibody at day 0. We then split the subjects into two groups whose Day 0 antibody magnitude was above or below this median. The average rise in spike log₁₀ concentration (BAU/ml) was 0.53 for the low group and 0.07 for the high group (p=0.015 using a t-test). Results were similar for neutralization titer with the average log₁₀ ID₅₀ titer rising 0.76 for the low group and 0.13 for the high group (p=0.018 using a t-test). Thus, for higher titer vaccinated individuals, symptomatic infection can be cleared without invoking a measurable anamnestic response. We speculate that for asymptomatic infections that do not develop into COVID-19 an anamnestic response may be important at lower tiers but not at higher titers”

4) All analyses and results apply to the pre-Omicron era (as briefly stated in the section on study limitations). Neutralization titers against Omicron after two mRNA vaccine doses are typically very low. After post-vaccination breakthrough infection, a strong increase in neutralizing serum titers against Omicron has repeatedly been demonstrated (recall response, including for breakthrough infections with a non-Omicron variant). Nevertheless, vaccine effectiveness against Omicron-mediated symptomatic COVID-19 after two mRNA vaccine doses is very low. Thus, symptomatic infection can frequently occur in vaccinated individuals with low neutralization titers despite a nAb recall response. This aspect should be highlighted more. This is especially true as non-nAb-based mechanisms of protection are likely not as affected by Omicron mutations, even more suggesting a role for high antibody titers in preventing symptomatic COVID-19.

We agree, our analysis was restricted to the mostly ancestral and pre-Omicron era. For Omicron, titers are lower and vaccine effectiveness lessened compared to the pre-Omicron era which might suggest that higher titers still prevent COVID-19. Since we do not have data on these points, we make the following point in the discussion (lines 360-361).

“Our studies were conducted in the pre-Omicron era limiting the generalizability of our results to additional variants. Ongoing work is assessing the impact of neutralization tier on Omicron disease and will be important to contrast with our results.”

5) In the section on study limitations, the authors note that testing asymptomatic infection in the COV-2069 trial was participant-driven. To my understanding, asymptomatic infections were not included in this manuscript as only protection from symptomatic COVID-19 cases was analyzed. This statement should perhaps be removed to avoid confusion.

We provide supportive PE curves for any infection and for asymptomatic infection in the supplement and describe them in the text. To clarify we added the following to the methods, line 108 when describing asymptomatic testing:

“We analyze asymptomatic infection in supportive analyses.”

Reviewers' Comments:

Reviewer #1:

Remarks to the Author:

Minor concerns

Deming regression requires an extra parameter - ratio of the errors. What was the value used in the analysis? Note - blue and red lines may not be clearly distinguished in Fig S2. Use dashing in one of the lines as a second marker to separate the lines.

When citing p values from likelihood ratio test, please also list the actual value for χ^2 distribution. This will allow to improve reproducibility.

Equation numbering in supplement is poorly formatted (too close to equations). If the journal edits the supplement, I assume that these will get fixed. However, some journals post supplement as it is, so perhaps making the equation numbering proper would be useful. A note - the first set of equations on how Ab dynamics was modeled are not numbered.

Reviewer #3:

Remarks to the Author:

I thank the authors for adequately addressing my comments.

Two final comments:

1)

I agree that circulating antibodies, both after monoclonal antibody administration and vaccination will penetrate to the mucosa. However, I am not certain whether the extent to which they do so in these settings can necessarily be compared.

Casirivimab and imdevimab are both IgG1 isotype antibodies (all neutralizing), whereas vaccination will induce different subclasses of antibodies (different IgG isotypes, IgM, and IgA), both neutralizing (quantifiable by neutralization assays) and non-neutralizing (not detected in neutralization assays). These different subclasses not only differ in their potential to penetrate to mucosal tissues but also in their non-neutralizing functions (e.g., ADCC, ADCP).

Thus, I feel that the deconstruction analysis grouping of "antibodies" (both serum and mucosal; including non-neutralizing functions performed by both neutralizing as well as non-neutralizing antibodies) vs. non-antibody functions based on serum neutralization may be somewhat limited by this, and this should perhaps be acknowledged.

2)

"These titers in IU50/ml can be converted to reciprocal dilution titers by dividing by 0.242. Thus, a titer of 1000 IU50/ml corresponds to an ID50 of 4132.2 ug/ml."

Titers (dilutions) are not expressed in $\mu\text{g/ml}$.

REVIEWERS' COMMENTS

Reviewer #1 (Remarks to the Author):

Minor concerns

Deming regression requires an extra parameter - ratio of the errors. What was the value used in the analysis? Note - blue and red lines may not be clearly distinguished in Fig S2. Use dashing in one of the lines as a second marker to separate the lines.

In the supplementary materials we now write.

Figure S2 displays a 45-degree line (red) and a fitted line using standard Deming regression (dashed blue) which assumes the ratio of the error variances for the two assays are equal.

We also edited the caption to Figure S2.

When citing p values from likelihood ratio test, please also list the actual value for χ^2 distribution. This will allow to improve reproducibility.

We now report the likelihood ratio statistics which are 10.12 with 2 degrees of freedom for the test of a non-null effect of titer on protective efficacy and 8.20 with 1 degree of freedom for the test of a non-null effect of titer on vaccine efficacy.

Equation numbering in supplement is poorly formatted (too close to equations). If the journal edits the supplement, I assume that these will get fixed. However, some journals post supplement as it is, so perhaps making the equation numbering proper would be useful. A note - the first set of equations on how Ab dynamics was modeled are not numbered.

Thanks, we've put the equation numbers to be right justified and added equation numbers for the first set of equations.

Reviewer #3 (Remarks to the Author):

I thank the authors for adequately addressing my comments.

Two final comments:

1)

I agree that circulating antibodies, both after monoclonal antibody administration and vaccination will penetrate to the mucosa. However, I am not certain whether the extent to which they do so in these settings can necessarily be compared.

Casirivimab and imdevimab are both IgG1 isotype antibodies (all neutralizing), whereas vaccination will induce different subclasses of antibodies (different IgG isotypes, IgM, and IgA), both neutralizing (quantifiable by neutralization assays) and non-neutralizing (not detected in neutralization assays). These different subclasses not only differ in their potential to penetrate to mucosal tissues but also in their non-neutralizing functions (e.g., ADCC, ADCP).

Thus, I feel that the deconstruction analysis grouping of “antibodies” (both serum and mucosal; including non-neutralizing functions performed by both neutralizing as well as non-neutralizing antibodies) vs. non-antibody functions based on serum neutralization may be somewhat limited by this, and this should perhaps be acknowledged.

In the limitations paragraph we now write.

In deconstructing the role of extant antibody in vaccine induced protection, we used casirivimab and imdevimab which are both IgG1 antibodies and are an imperfect proxy for mRNA-1273 induced antibodies which are polyclonal and include other IgG isotypes, IgM and IgA, which may differ in their potential to penetrate the mucosa and in non-neutralizing functions.

2)

“These titers in IU50/ml can be converted to reciprocal dilution titers by dividing by 0.242. Thus, a titer of 1000 IU50/ml corresponds to an ID50 of 4132.2 ug/ml.”

Titers (dilutions) are not expressed in µg/ml.

Thanks, deleted.